# Transcriptomic classes of *BCR-ABL1* lymphoblastic leukemia

Jaeseung C. Kim[1,2,3], Michelle Chan-Seng-Yue [1], Sabrina Ge [1,3], Andy G. X. Zeng[1], Karen Ng[1], Olga I. Gan[1], Laura Garcia-Prat[1], Eugenia Flores-Figueroa[1], Tristan Woo [1,3], Amy Xin Wei Zhang[2], Andrea Arruda[1], Shivapriya Chithambaram[1,2], Stephanie M. Dobson[1], Amanda Khoo[1,3], Shahbaz Khan[1], Narmin Ibrahimova[1], Ann George[1], Anne Tierens [1,3], Johann Hitzler [4], Thomas Kislinger [1,3], John E. Dick[1,3], John D. McPherson [2,5], Mark D. Minden[1,3] & Faiyaz Notta [1,2,3] ✉

In *BCR-ABL1* lymphoblastic leukemia, treatment heterogeneity to tyrosine kinase inhibitors (TKIs), especially in the absence of kinase domain mutations in *BCR-ABL1*, is poorly understood. Through deep molecular profiling, we uncovered three transcriptomic subtypes of *BCR-ABL1* lymphoblastic leukemia, each representing a maturation arrest at a stage of B-cell progenitor differentiation. An earlier arrest was associated with lineage promiscuity, treatment refractoriness and poor patient outcomes. A later arrest was associated with lineage fidelity, durable leukemia remissions and improved patient outcomes. Each maturation arrest was marked by specific genomic events that control different transition points in B-cell development. Interestingly, these events were absent in *BCR-ABL1*⁺ preleukemic stem cells isolated from patients regardless of subtype, which supports that transcriptomic phenotypes are determined downstream of the leukemia-initialing event. Overall, our data indicate that treatment response and TKI efficacy are unexpected outcomes of the differentiation stage at which this leukemia transforms.

The *BCR-ABL1* fusion drives two hematopoietic malignancies—chronic myelogenous leukemia (CML), a myeloproliferative disorder, and *BCR-ABL1* lymphoblastic leukemia, an acute leukemia mostly of the B-cell lineage. Despite sharing the same initiating event, patients with these diseases show vastly different outcomes. Most CML patients can expect to live a near-normal lifespan owing to targeted therapy with tyrosine kinase inhibitors (TKIs), whereas the survival of *BCR-ABL1* lymphoblastic leukemia patients remains dismal despite TKI intervention[1,2]. Approximately half of the patients diagnosed with *BCR-ABL1* lymphoblastic leukemia die within 5 years of being diagnosed[2]. Most patients relapse due to the emergence of clones with kinase domain mutations, but 30–40% of patients relapse without kinase domain

mutations, which is poorly understood[3]. TKI resistance is mostly viewed through the lens of kinase domain mutations, but whether kinase domain-independent mechanisms of therapy resistance contribute to the emergence of kinase domain-mutant clones is unknown.

The genomics of *BCR-ABL1* lymphoblastic leukemia is well described. Loss of *IKZF1* is present in 80% of patients[4] and mouse models have shown that *IKZF1* alterations are linked to TKI resistance[5]. As many patients with *BCR-ABL1* lymphoblastic leukemia demonstrate durable responses to treatment, some of them must harbor *IKZF1* losses given the high frequency of this alteration. This discordance remains unexplained. Genetic alterations in other B-cell differentiation genes, such as *PAX5*, *CDKN2A/B* and *EBF1*, are also frequent in this disease[4].

[1]Princess Margaret Cancer Centre, Toronto, Ontario, Canada. [2]Ontario Institute for Cancer Research, Toronto, Ontario, Canada. [3]Department of Medical Biophysics, University of Toronto, Toronto, Ontario, Canada. [4]The Hospital for Sick Children, Toronto, Ontario, Canada. [5]University of California Davis Comprehensive Cancer Center, Sacramento, CA, USA. ✉e-mail: faiyaz.notta@gmail.com

The co-occurrence of multiple driver alterations makes it difficult to decipher their roles in the heterogeneity of treatment outcomes among patients. Furthermore, this disease presents two drastically different molecular phenotypes–mixed-lineage and classic B-cell precursor. These phenotypes are thought to reflect the differentiation potential of the cell-of-origin, but this remains controversial. There are two major isoforms of *BCR-ABL1*. The long isoform (p210) has been shown to originate in a hematopoietic stem cell (HSC), whereas the short isoform (p190) has been shown to originate in a B-cell progenitor[6]. Rationally, this leads to the notion that mixed-lineage phenotype is related to the p210 isoform originating in an HSC, whereas the classic B-cell phenotype is related to the p190 isoform originating in a committed B-cell progenitor. However, recent work tracking residual disease in patients shows that p190 isoform can be sustained in HSC but can produce acute leukemia with a typical B-cell phenotype. The origins of these disease phenotypes and how they influence treatment outcomes are not well understood[7].

To address the issues raised above, we collected a cohort of *BCR-ABL1* lymphoblastic leukemia patient samples with extensive treatment response data. Our data propose a model of pathogenesis that explains the molecular and clinical heterogeneities observed in this disease.

## Results

### Distinct transcriptomic clusters of *BCR-ABL1* lymphoblastic leukemia

Two patient cohorts spanning 20 years (1992–2019) with clinical follow-up data were collected (Extended Data Fig. 1a). The first cohort of 57 samples from 53 patients (Supplementary Table 1; 46 de novo *BCR-ABL1* B-cell acute lymphoblastic leukemia (B-ALL), 5 CML in lymphoid blast crisis and 2 mixed phenotype acute leukemia (MPAL)) was purified for blasts and subjected to RNA sequencing (RNA-seq; Extended Data Fig. 1b). Non-negative matrix factorization (NMF) followed by consensus clustering uncovered three distinct transcriptomic clusters, initially labeled C1, C2 and C3 (Fig. 1a, Supplementary Figs. 1 and 2, Extended Data Figs. 1c and 2 and Supplementary Table 2). These clusters were replicated in the second cohort of 40 patients (Supplementary Figs. 3 and 4 and Supplementary Table 3). In the combined cohort (97 samples, 93 patients), the proportions of C1, C2 and C3 were 40%, 22% and 38%, respectively. These clusters were not associated with age, sex, sample source (blood or bone marrow (BM)) or *BCR-ABL1* isoform (p190 or p210); white blood cell counts at diagnosis were greater in C1 compared with C2 and C3 (Extended Data Fig. 3 and Supplementary Table 4). We also found these clusters in Ph-like ALL, suggesting they may apply to other forms of ALL (Supplementary Fig. 5).

We first used gene set enrichment analysis (GSEA) to examine these transcriptomic clusters. C1 was marked by the aberrant expression of stem and myeloid lineage genes (for example *KIT*, *CSF3R* and *MECOM*; Fig. 1b) and was enriched for pathways related to innate immunity (TNFα, inflammatory response; Supplementary Fig. 6, top). C2 showed some expression of both myeloid (*CSF2RA* and *CSF1R*) and lymphoid genes (*MS4A1/CD20* and *IL7R*) and was enriched for pathways related to ERK signaling, unfolded protein response (UPR) and other pathways not related to hematopoiesis (myogenesis, epithelial to mesenchymal transition (EMT); Supplementary Fig. 6, middle). C3 showed the highest expression of genes associated with B-cell differentiation, such as *IL7R*, *MS4A1*, *BACH2* and *TCL1A*, and was strongly enriched for cell cycle pathways (E2F targets, G2M checkpoint) and B-cell receptor signaling (Supplementary Fig. 6, bottom).

We then compared these transcriptomic clusters to flow cytometry data. Eight antigens were differentially expressed between the clusters (*P* = 0.0059–0.046, Kruskal–Wallis test; Fig. 1c and Supplementary Fig. 7a–d). Myeloid antigens, CD13, CD33 and cytoplasmic myeloperoxidase (cyMPO), and a T-cell antigen, CD7, were expressed at the highest levels in C1, whereas B-cell antigens, CD10, CD19 and CD20,

were all expressed at the highest levels in C3. Notably, CD20, a mature B-cell antigen, was often absent in C1 and C2 but frequently expressed in C3. This may have implications for targeted therapy using monoclonal CD20 antibodies, such as rituximab[8]. C1 cluster also showed the highest expression of the stem cell antigen, CD34 (*P* = 0.18, Kruskal–Wallis test; Supplementary Fig. 7b). C1 frequently expressed both myeloid and lymphoid antigens, whereas C3 displayed more restricted expression of lymphoid antigens (further analysis is given in Supplementary Note). Surface antigen expression and gene expression of lineage markers closely mirrored each other (Supplementary Fig. 8), indicating that transcriptomic clusters associate with surface phenotypes.

MPALs are enriched for the *BCR-ABL1* fusion[9,10]. Because C1 leukemias co-expressed lymphoid and myeloid antigens, we investigated why these were not diagnosed as MPAL. Ninety-seven percent of C1 were diagnosed as either B-ALL (*n* = 32/36, 89%) or CML-LBC (*n* = 3/36, 8%). Despite the expression of surface myeloid antigens, levels of cyMPO were below the diagnostic threshold of 10% for MPAL[11]. However, cyMPO levels were still significantly greater in C1 than in C2 and C3 (median 2% versus 0%; Fig. 1c and Supplementary Table 5). RNA-seq showed a greater dynamic range of detecting low levels of *MPO* compared with flow cytometry (Supplementary Fig. 7e)[12]. It is possible that current diagnostic guidelines may underestimate the number of *BCR-ABL1* lymphoblastic leukemia with mixed-lineage phenotype. Notably, despite the large number of gene expression studies in ALL, our focused approach identifies distinct transcriptomic subtypes of the disease.

### Transcriptomic clusters align with B-cell differentiation stages

We compared the transcriptomic clusters to their normal counterparts in human cord blood including HSCs, multipotent progenitors (MPPs), myeloid progenitors (common myeloid progenitors (CMPs), megakaryocyte–erythroid progenitors (MEPs), granulocyte/macrophage progenitors (GMPs)) and eight different lymphoid and B-cell progenitors (lymphoid-primed MPPs (LMPPs), multilymphoid progenitors (MLPs), early pro-B, pro-B, pre-B I, pre-B II, immature B and mature B; Supplementary Table 6 and Supplementary Fig. 9a,b). Principal component analysis (PCA) revealed that C1 leukemias are closely associated with early pro-B cells (Fig. 2a), despite their strong stem/myeloid gene expression pattern. Early pro-B is the first step in B-cell development and retains myeloid and T-lymphoid lineage potentials[13]. C2 and C3 leukemias are associated with pro-B and pre-B I cells, which are progenitors fully committed to the B-cell lineage[14]. To further delineate the degree of differentiation in C2 and C3 leukemias, we assessed the expression of immunoglobulin heavy and light chains. Expression of rearranged heavy chain (*IGH*) was similar across all clusters (Supplementary Fig. 10a), whereas expression of rearranged light chains (*IGK* or *IGL*) was most frequent in C3 (Fig. 2b). C3 leukemias also showed marked upregulation of B-cell transcription factors, *EBF1* and *PAX5*, and B-cell receptor signaling genes (Fig. 2c,d). This suggests a later differentiation block in C3 compared with C2. These analyses support that C1, C2 and C3 leukemias are blocked at different stages of early B-cell development. Similar findings were observed from human adult BM progenitors[15] (Supplementary Fig. 9c). Taken together, we labeled the transcriptomic subtypes as follows: C1, early B-cell progenitor ('Early-Pro'); C2, intermediate B-cell progenitor ('Inter-Pro'); and C3, late B-cell progenitor ('Late-Pro'; Fig. 2e).

### Early-Pro leukemias display significant hematopoietic lineage plasticity

We then applied single-cell RNA-seq (scRNA-seq) to nine samples (four Early-Pro, one Inter-Pro and four Late-Pro; *n* = 29,411 total single cells). Nonleukemic cells were removed based on lineage marker genes (for example, *DNTT* and *CD19*; Methods). SingleR was used to match individual leukemic cells to their closest normal cell counterparts[16].

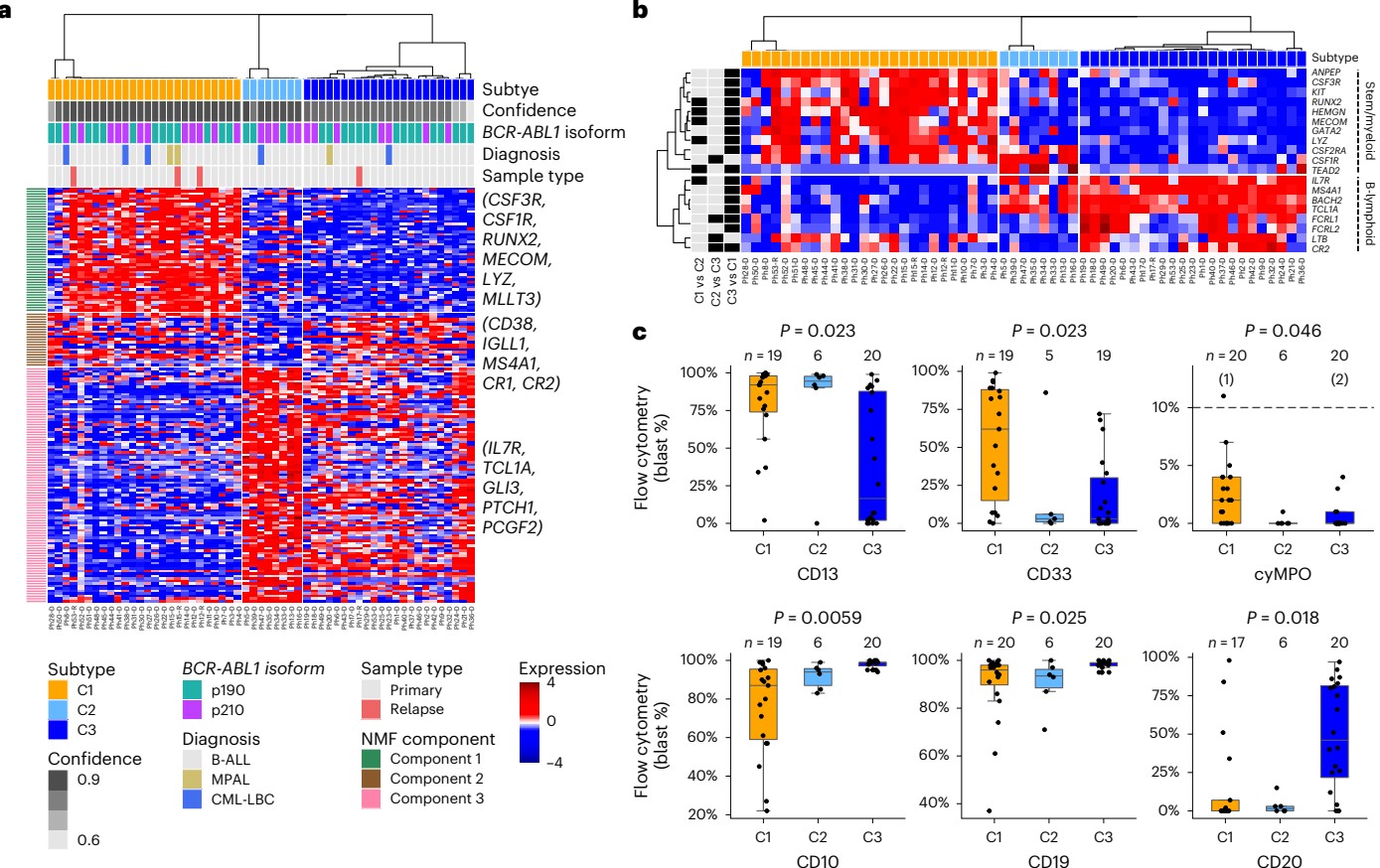

**Fig. 1 | Three molecular subtypes of *BCR-ABL1* lymphoblastic leukemia.** **a**, Gene expression heatmap of three molecular subtypes in *BCR-ABL1* lymphoblastic leukemia identified by consensus hierarchical clustering. Molecular subtypes, clustering confidence scores, *BCR-ABL1* isoforms, clinical diagnoses and sample types are shown in tracks. A subset of hematopoietic lineage genes in each NMF component is shown. **b**, Heatmap of selected differentially expressed genes (FDR-adjusted *P* < 0.05 from pairwise comparison of subtypes) in stem/myeloid and B-lymphoid programs. Sample order and heatmap scale are the same as in **a**. Tracks on the left display from which pairwise comparison(s) each gene was derived. **c**, Proportions of blasts positive for myeloid (top) and B-lymphoid (bottom) lineage antigens by subtype. For cyMPO, the dashed line represents the diagnostic threshold of 10% and numbers in brackets represent outliers. Counts represent the number of primary leukemias assessed for the antigen. FDR-adjusted *P* values from Kruskal–Wallis tests are shown. In each boxplot in this paper, the middle line represents the median, the lower and upper edges of the box represent the first and third quartiles, the end of the lower whisker represents the smallest value at most 1.5× interquartile range from the lower edge of the box, the end of the upper whisker represents the largest value at most 1.5× interquartile range from the upper edge of the box and all data points were shown.

In the four Late-Pro samples, 95.3% of leukemic cells were assigned as either pro-B (46.0 ± 7.3%) or pre-B I cells (48.8 ± 3.7%) (Fig. 2f and Extended Data Fig. 4a,b). The Inter-Pro sample showed decreased assignment to pro-B (36.6%) and pre-B I (40.5%) cells and increased assignment to early pro-B cells (21.4%), consistent with an earlier block in differentiation than Late-Pro leukemias. Early-Pro samples showed significantly more lineage heterogeneity compared with Inter-Pro and Late-Pro samples. They were composed of more early pro-B cells (40.4 ± 7.5%) and fewer pro-B (8.8 ± 5.7%) and pre-B I cells (11.0 ± 3.2%) than the other two subtypes (Fig. 2g). Early-Pro was the only subtype where we observed transcriptional similarity to earlier stages of lymphoid development, such as LMPP (6.0 ± 1.4%) and MLP (0.7 ± 0.7%), as well as GMP (23.6 ± 9.7%). Interestingly, GMPs in both mice and humans show lymphoid potential[17,18]. The Shannon indices of Early-Pro leukemias were significantly greater than those of Late-Pro leukemias (median of 1.49 versus 0.76; *P* = 0.029, Wilcoxon rank-sum test; Extended Data Fig. 4c) confirming that they are more heterogeneous. Similar results were obtained when using adult BM[19] (Supplementary Fig. 11a–c). Pseudotime analysis of Early-Pro samples showed GMP-like and pro-B/pre-B I-like cells emerge from early pro-B-like cells, suggesting that mixed-lineage program originates from a lymphoid state (Supplementary Fig. 11d). Together, Early-Pro

leukemias show considerable lymphomyeloid lineage plasticity compared with Inter-Pro and Late-Pro leukemias.

**Distinct genetic alterations define each molecular subtype**
Whole-genome sequencing (WGS) was performed on 57 patients from the first cohort. Known recurrent alterations were found in *IKZF1* (77%), *PAX5* (42%), *CDKN2A/B* (38%), *SLX4IP* (25%), *HBS1L* (23%), *BTG1* (19%), *RB1* (15%), *EBF1* (15%), *MEF2C* (15%), *RUNX1* (15%) and others (Fig. 3e). New recurrent alterations were found in *CBWD2* (36%), *GPN3/FAM216A* (13%), *PRKAR2B* (11%; excluding monosomy 7 cases), *MAP3K2* (5.7%), *MIR181A1HG/MIR181B1* (5.7%) and *FOXP1* (5.7%). Notably, 96.2% of leukemias (*n* = 51/53) harbored alterations in genes that regulate normal lymphoid differentiation (*IKZF1*, *PAX5*, *BTG1*, *RUNX1*, *EBF1*, *MEF2C*, *TSC22D1*, *FOXP1*, *LEF1*, *ETV6*, *TCF3* and *TCF4*; mean of 2.1 alterations per leukemia; Extended Data Fig. 5a,b) and consistent with previous findings[4]. Thus, a limited set of transformation events cooperate with *BCR-ABL1* to disrupt lymphoid development.

Among the transcriptomic subtypes, *EBF1* alterations were restricted to Early-Pro leukemias (*n* = 8/23, 35%; *P* = 0.0018, Fisher's exact test is used throughout this section; Fig. 3a). In the initial steps of B-cell commitment, *EBF1* represses myeloid and T-cell lineages[20] and loss of *EBF1* would derepress these lineage programs, as found in

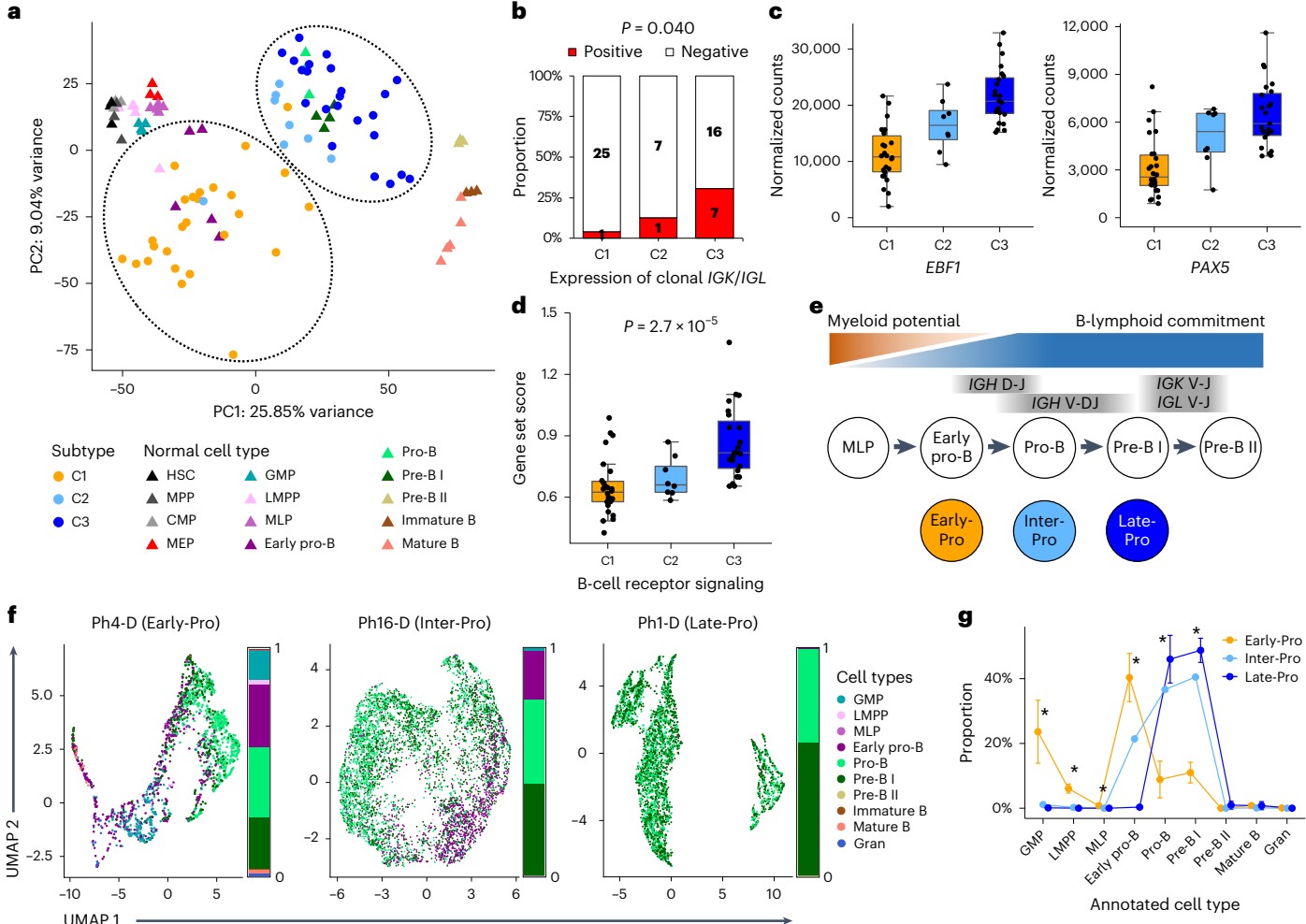

**Fig. 2 | Molecular subtypes are arrested at distinct stages of B-cell differentiation. a**, PCA of *BCR-ABL1* lymphoblastic leukemias (circles) and normal hematopoietic cell types (triangles) using 2,000 most variable genes. **b**, Proportions of leukemias in each subtype with and without expression of clonally rearranged immunoglobulin light chain genes (*IGK*, *IGL*). *P* value is from Fisher's exact test. **c**, Normalized counts of B-cell development genes, *EBF1* and *PAX5*, by subtype (26 C1, 8 C2 and 23 C3 leukemias). **d**, Gene set score of B-cell receptor signaling by subtype (26 C1, 8 C2 and 23 C3 leukemias). **e**, Approximate positioning of the molecular subtypes in relation to B-cell differentiation stages. Transition from early pro-B to pro-B is accompanied by decreasing myeloid

potential and increasing commitment to the B-lymphoid program. Timings of immunoglobulin heavy chain (*IGH*) and light chain (*IGK*, *IGL*) rearrangements are also shown. **f**, Single cells from scRNA-seq are annotated with their closest normal cell counterparts and visualized using UMAP. Exemplar cases from three subtypes are shown. Bars represent the proportions of cell type annotations. **g**, Proportions (median with interquartile range) of cell type annotations in each subtype (four Early-Pro, one Inter-Pro and four Late-Pro). Asterisks represent FDR-adjusted *P* < 0.05 from Wilcoxon rank-sum tests comparing Early-Pro and Late-Pro leukemias.

Early-Pro leukemias. Alterations in *RUNX1* (*AML1*) were also restricted to Early-Pro leukemias (*n* = 8/23, 35%; *P* = 0.0018). *RUNX1* is required for the activation of *EBF1* and is crucial for lymphoid specification[21]. *HBS1L* deletions were also only found in Early-Pro leukemias (*n* = 12/23, 52%; *P* = 1.5 × 10⁻⁵). Monosomy 7, a cytogenetic aberration in myeloid leukemias[22,23], was enriched in Early-Pro cases (*n* = 7/23, 30%; *P* = 0.076). Overall, Early-Pro leukemias harbor alterations that deregulate early steps of B-lymphoid commitment.

Although *IKZF1* alterations were present among all transcriptomic subtypes, bi-allelic losses were significantly enriched in Inter-Pro leukemias (*n* = 7/8, 88%; *P* = 2.0 × 10⁻⁷; Fig. 3a and Extended Data Fig. 5c). Inter-Pro leukemias were also enriched for the dominant negative *Ik6* deletion of *IKZF1*, which lacks the DNA binding domain (62%, 36% and 17% in Inter-Pro, Late-Pro and Early-Pro, respectively; *P* = 0.012; Extended Data Fig. 5d). Inactivation of *IKZF1* activates stromal genes[5], and accordingly, Inter-Pro samples were significantly enriched for aberrant expression of these genes (Extended Data Fig. 5e). Interestingly,

one Late-Pro leukemia, Ph36-D, harbored a dominant negative mutation in *IKZF1* (p.N159T)[24] and showed a similar pattern of stromal cell disruption and clustered close to Inter-Pro leukemias (Supplementary Fig. 12). This suggests that complete inactivation of *IKZF1* drives the Inter-Pro subtype.

Late-Pro cases were enriched for *PAX5* deletions (*n* = 15/22, 68%; *P* = 0.0022). *PAX5* is activated downstream of *EBF1* and facilitates pro-B to the pre-B cell transition, immunoglobulin gene rearrangements and pre-B cell receptor signaling[25,26]. Increased frequency of *PAX5* deletions is consistent with a later block in B-cell differentiation in Late-Pro leukemias. Late-Pro leukemias were also significantly enriched for homozygous deletions of *CDKN2A/B* and *RB1* (*P* = 0.0013 and 0.0047). *CDKN2A/B* locus was often lost together with *PAX5* as part of chromosome 9p losses. *CDKN2A/B* and *RB1* regulate the pre-B cell receptor checkpoint to ensure cells with nonproductive *IGH* rearrangements undergo cell cycle arrest and apoptosis[27]. Together, losses of *CDKN2A/B*, *RB1* and *PAX5* reconcile how Late-Pro leukemias escape the pre-B cell

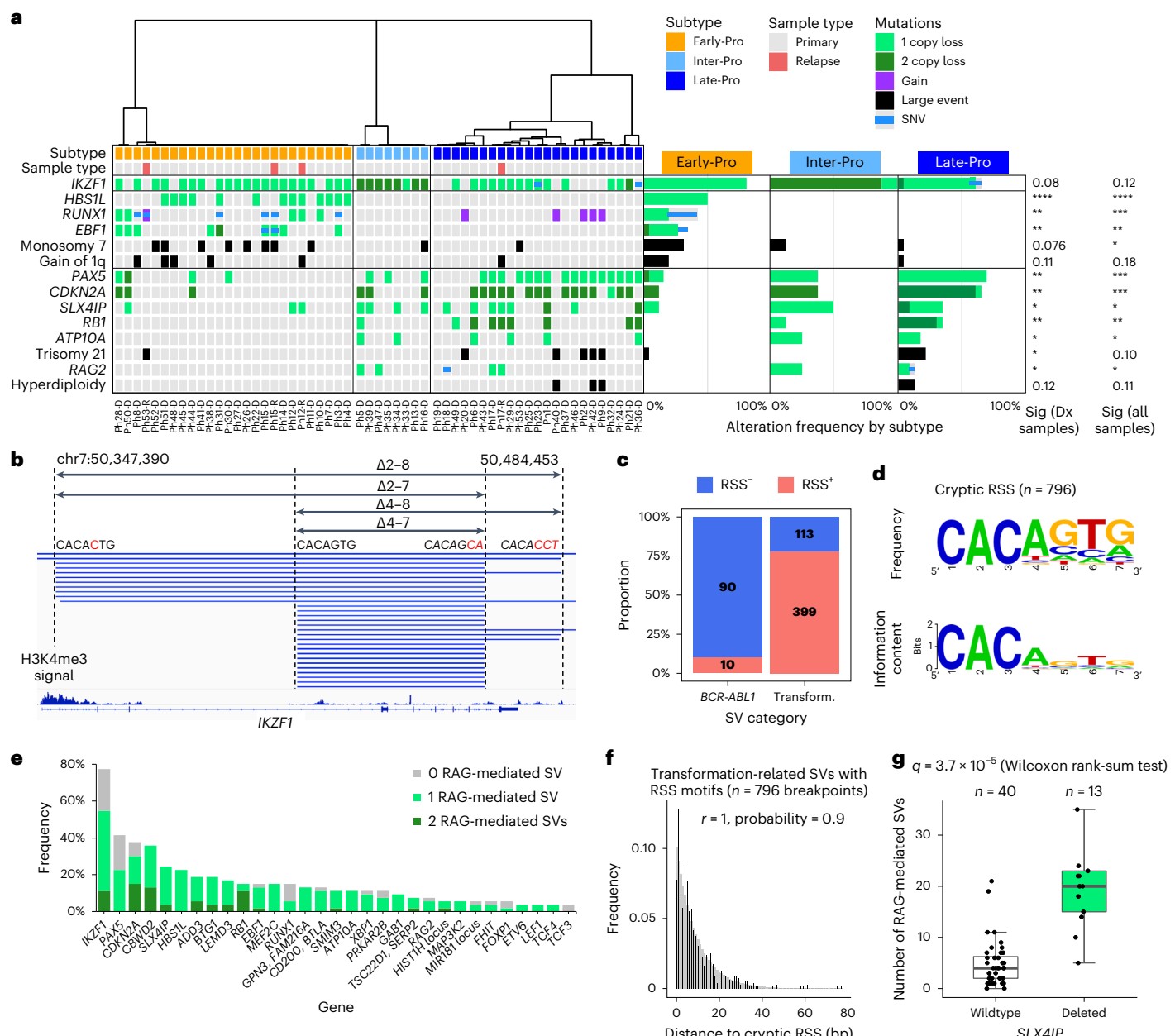

**Fig. 3 | Distinct cooperating genetic alterations define each molecular subtype. a**, Oncoprint of genetic alterations enriched in each subtype. Samples are in the same order as in Fig. 1a. Molecular subtype, sample type (diagnosis/relapse) and gene mutations are shown for each sample. Alteration frequencies in each subtype are shown on the right. *IKZF1* deletions include monosomy 7. Copy number gains from trisomy 21 are not included in *RUNX1* alteration frequencies. In hyperdiploid cases, trisomy 21 is a consequence of the hyperdiploid state. Fisher's exact test was performed using only diagnostic samples or all samples. **b**, Recurrent *IKZF1* deletions (blue lines) are generated by RAG-mediated recombination. RSS motifs near breakpoint clusters are shown. Right-side RSS are written in reverse complement and italicized. Bases that deviate from the

canonical RSS ('CACAGTG') are in red. H3K4me3 ChIP–seq signal for GM12878 (B-lymphoblast cell line) from ENCODE is shown at the bottom. **c**, Proportions of *BCR-ABL1*-associated and transformation-related (Transform.) SVs with and without RSS motifs. **d**, Sequence logo of cryptic RSS heptamers from transformation-related SVs generated using WebLogo[48]. **e**, Frequencies of leukemias with alterations in each gene. Colors correspond to 0, 1 or 2 RAG-mediated SVs. **f**, Distances between SV breakpoints and nearest RSS motifs (black) in transformation-related SVs form a negative binomial distribution (gray). **g**, Numbers of RAG-mediated recombinations in primary leukemias by deletion status of *SLX4IP*. Sig, statistical significance; Dx, diagnosis. ****$P < 5 \times 10^{-5}$ < ***$P < 5 \times 10^{-4}$ < **$P < 0.005$ < *$P < 0.05$.

receptor checkpoint and maintain a proliferative state (Supplementary Fig. 6, bottom, and Extended Data Fig. 5f)[25]. Two Early-Pro cases, Ph28-D and Ph50-D, harbored concurrent deletions of *EBF1*, *PAX5*, *RUNX1* and *CDKN2A/B* and clustered between Early-Pro and Late-Pro leukemias in PCA (Supplementary Fig. 12). With regards to large-scale copy number aberrations, trisomy 21 was moderately enriched in Late-Pro leukemias ($n = 5/22$, 23%; $P = 0.020$; Supplementary Fig. 13) and was found to

be mutually exclusive with *IKZF1* losses (FDR-adjusted $P = 0.039$; Supplementary Table 7). Hyperdiploidy was also only observed in Late-Pro leukemias ($n = 3/22$, 14%; $P = 0.12$). Mutation loads of single-nucleotide variants (SNVs) and indels were similar across all transcription subtypes (Extended Data Fig. 5g). These patterns of genetic alterations suggest that each transcriptomic subtype transforms at a distinct stage of B-cell development.

## DNA footprints of lymphoid enzymes inform on the timing of transformation

Recombination-activating gene (RAG) endonuclease (RAG1/2) introduces DNA double-strand breaks during V(D)J recombination in lymphoid progenitors at a specific motif known as the recombination signal sequences (RSS; Extended Data Fig. 6a)[28]. The presence of the RSS motifs at a structural variant (SV) outside of the antigen receptor genes indicates off-target RAG activity and is common in lymphoid leukemias[29]. From the WGS data, 632 SVs were resolved at the nucleotide level for RSS motif analysis (Supplementary Table 8; Methods). Nineteen percent of these SVs ($n = 120/632$) were associated with *BCR-ABL1* translocations, and the remaining 81% ($n = 512/632$) were associated with cooperating events (for example, deletions in *IKZF1*, *PAX5* and *CDKN2A/B*; Fig. 3b and Supplementary Fig. 15). SVs related to cooperating events were significantly enriched for the RSS motif compared with SVs from *BCR-ABL1* translocations (78% versus 8%; $P < 1 \times 10^{-16}$, Fisher's exact test; Fig. 3c–e). Only SVs related to cooperating events showed a decline in breakpoint frequency as a function of distance from the RSS motif and formed a negative binomial distribution (Fig. 3f and Extended Data Fig. 6b,c). Interestingly, leukemias with deletions of *SLX4IP* harbored significantly more RAG-mediated SVs (Fig. 3g and Supplementary Note). Early-Pro, Inter-Pro and Late-Pro leukemias showed similar numbers of RAG-mediated SVs (Extended Data Fig. 6d), supporting that this mutational process drives transformation in all transcriptomic subtypes.

Terminal deoxynucleotidyl transferase (TdT), encoded by *DNTT*, is another key enzyme in V(D)J recombination. Similar to *RAG1/2* genes, *DNTT* is expressed in early lymphoid progenitors but is absent in multipotent cells such as HSC. TdT introduces nontemplate sequences (NTS) during repair of double-strand breaks. NTS insertions (mean length of 5.58 nucleotides) were detected in 94.2% of SVs related to cooperating events with RSS motifs ($n = 376/399$; Supplementary Table 8 and Supplementary Fig. 16a,b). This is consistent with RAG-mediated recombination in B-cell progenitors. Approximately two-thirds of the inserted nucleotides were G or C, reflecting the preferential usage of G:C over A:T bases by TdT[30] (Supplementary Fig. 16c). Interestingly, 65.5% of SVs related to cooperating events without RSS motifs ($n = 74/113$) also showed NTS insertions with a similar mean length (5.93 nucleotides). This suggests that TdT was active at SVs not created by RAG recombination. For SVs associated with *BCR-ABL1* translocation, 91.8% ($n = 109/120$) did not harbor an NTS. The mean length of NTS for the small fraction of *BCR-ABL1*-associated SVs with NTS (9.2%; $n = 11/120$) was 2.55 nucleotides (Supplementary Fig. 16a,b). This supports that TdT was not active when the *BCR-ABL1* translocation occurred. To summarize, SVs related to cooperating events in the transcriptomic subtypes carried DNA footprints of lymphoid enzyme activity, which suggests the cooperating events defining each subtype were conceived in a lymphoid cell downstream of *BCR-ABL1*.

### *BCR-ABL1* translocation arises independently and upstream of cooperating events

To investigate this notion, hematopoietic stem and progenitor cells (CD34+ CD19− CD45RA−) from 14 patients (six Early-Pro, two Inter-Pro and six Late-Pro) were sorted and subjected to colony formation assay, an assay that removes residual blasts and promotes stem/progenitor differentiation into myeloid cells (Fig. 4a, Extended Data Fig. 7a and Supplementary Note). Colonies ($n = 771$; 21–87 per sample) were picked and analyzed for patient-specific genomic alterations using nested PCR (Extended Data Fig. 7b and Supplementary Table 9). *BCR-ABL1* translocations were detected in granulocyte/macrophage (GM) and erythroid colonies in 4 of 14 patients (29%; three p210 and one p190; Fig. 4b, Extended Data Fig. 7c and Supplementary Table 10), a frequency consistent with previous studies[7,31]. We expected more Early-Pro cases to harbor *BCR-ABL1*+ colonies given that mixed-lineage leukemias have been shown to arise from HSC[32]. Interestingly, leukemias with

*BCR-ABL1*+ preleukemic clones were not restricted to the Early-Pro subtype ($P = 0.78$, Fisher's exact test). No myeloid colonies with the *BCR-ABL1* translocation carried cooperating events, suggesting that the *BCR-ABL1* translocation likely occurs in a multilineage progenitor (HSC/MPP) before lymphoid commitment. This is in accordance with previous reports of detecting rare *BCR-ABL1*+ clones in blood from healthy individuals[33,34]. Although our numbers are small, we did not find evidence that the mixed-lineage phenotype of Early-Pro subtype is due to the transformation of a multipotent cell.

Four patients had paired samples from diagnosis and relapse. Three had relapsed within 2 years from diagnosis (two Early-Pro and one Late-Pro; Fig. 4c). In these patients, most variants present at diagnosis were found at relapse (Fig. 4d and Supplementary Table 11). The transcriptomic subtypes, immunophenotypes and antigen receptor rearrangements also remained unchanged between diagnosis and relapse (Fig. 4e and Supplementary Figs. 17 and 18). This indicates that relapse emerged from the major diagnostic clone in these three patients. The fourth patient (Ph53) relapsed after a 4-year remission. Although the same *BCR-ABL1* translocation was shared, only 14% of the diagnostic mutations were present at relapse (Fig. 4d, bottom, and Extended Data Fig. 8a,d). Relapse was not driven by a kinase domain mutation in *BCR-ABL1*. There were significant changes in immunophenotypes and antigen receptor loci at relapse, indicating a new clone had emerged (Supplementary Figs. 17d and 18d and Supplementary Table 12). From RNA-seq, the diagnostic leukemia was classified as Late-Pro, whereas the relapse leukemia was classified as Early-Pro, indicating a switch in transcriptomic subtype (Fig. 4e). Genomic analysis showed an inactivating SV in *PAX5* at diagnosis, consistent with its Late-Pro subtype, but it was not detected at relapse (Fig. 4f and Extended Data Fig. 8d). Instead, the relapse sample harbored a *RUNX1* mutation consistent with the Early-Pro subtype (Fig. 4g and Extended Data Fig. 8e). This suggests a *BCR-ABL1*+ precursor clone, which lacked *PAX5* inactivation and *RUNX1* mutation, independently gave rise to both Late-Pro leukemia at diagnosis and Early-Pro leukemia at relapse (Supplementary Fig. 19). Together with the colony data, this supports that *BCR-ABL1* and cooperating events defining the transcriptomic subtypes occur at different stages of hematopoietic differentiation. Thus, the leukemic cell-of-origin does not define the transcriptomic subtype.

### Molecular subtypes impact treatment outcomes in patients

We obtained detailed patient response data for our cohort. Twelve of 93 patients were excluded from this analysis (five CML-LBC, five not given TKI and two died within 30 d). The remaining 81 patients received a mostly uniform frontline regimen of Dana Farber Cancer Institute (DFCI) protocol plus imatinib[35] (Supplementary Tables 1 and 3). Similar proportions of Early-Pro, Inter-Pro and Late-Pro patients (32–39%) received bone marrow transplants (BMTs). Residual disease was monitored by *BCR-ABL1* transcript levels with a median follow-up time of 6.3 years (95% confidence interval (CI): 4.9–10.1 years).

A major molecular response (MMR) is defined as ≥3 log reduction (≤0.1% transcript) in residual disease (Fig. 5a). Significantly fewer Early-Pro (10%, $n = 2/20$) and Inter-Pro patients (17%, $n = 2/12$) achieved MMR compared with Late-Pro patients (56%, $n = 14/25$) during induction therapy ($P = 0.0021$, Fisher's exact test). Median log reductions after induction for Early-Pro, Inter-Pro and Late-Pro patients were 1.4, 2.2 and 3.5, respectively ($P = 0.022$, Kruskal–Wallis test; Fig. 5b). A deep molecular response (DMR), defined as ≥4 log reduction, is a predictor of good outcome[36] and was restricted to Late-Pro patients (Fig. 5b). Accordingly, overall survival (OS) and event-free survival (EFS) rates were considerably greater in the Late-Pro subtype ($P = 0.019$ and 0.015, respectively; Fig. 5c). OS rate for Late-Pro patients was 73%, analogous to patients with low-risk B-ALL[37]. By contrast, Early-Pro and Inter-Pro patients had worse OR rates of 33% and 54%, respectively. Notably, Late-Pro patients showed good prognosis despite frequent single-copy losses or mutations in *IKZF1* ($n = 14/22$, 64%), suggesting single-copy

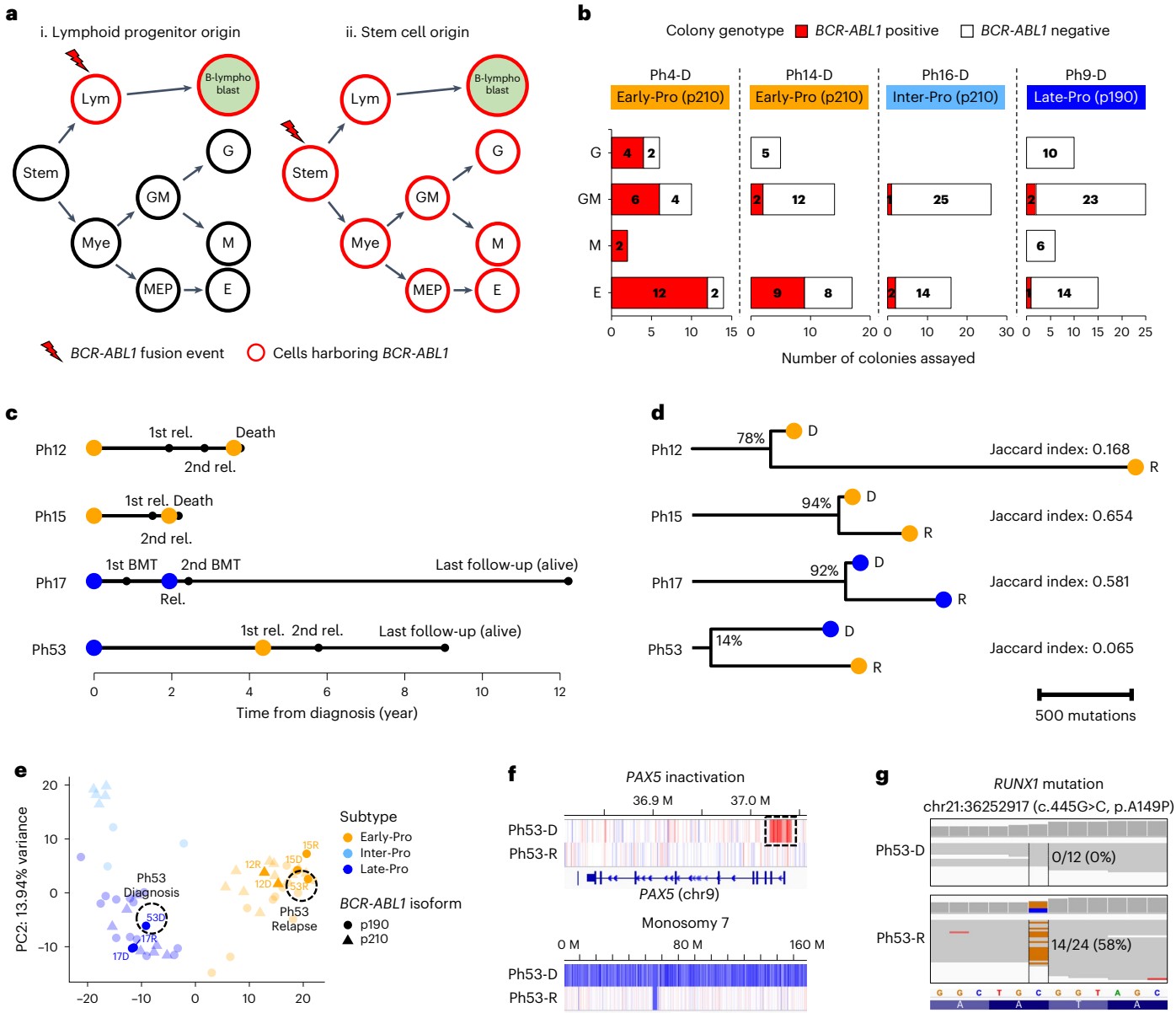

**Fig. 4 | Origin of *BCR-ABL1* lymphoblastic leukemia in a multipotent cell.**
**a**, Simplified lineage hierarchy of B-lymphoblast and four colony types. Two potential scenarios for the origin of *BCR-ABL1* are shown as follows: lymphoid progenitor origin (i) and stem cell origin (ii). Red outlines denote cell types that can harbor *BCR-ABL1* under each scenario. **b**, Four patient samples with *BCR-ABL1* positive colonies. For each colony type, numbers of *BCR-ABL1* positive and negative colonies are shown. **c**, Timelines of clinical events in four patients with diagnosis and relapse leukemia pairs. Two circles correspond to the times of two leukemia biopsies, and circle colors represent molecular subtypes, Early-Pro (orange) and Late-Pro (blue). **d**, Phylogenetic relationships of four leukemia pairs based on point mutations. Lengths of trunk (shared) and branches (private) are proportional to number of mutations. Percentage of mutations in diagnostic leukemia that are shared by relapse leukemia is shown.

The Jaccard index measures the degree of similarity between pairs[49]. **e**, PCA of leukemic transcriptomes using NMF component genes. Diagnosis/relapse pairs are labeled and highlighted. **f**, *PAX5* inactivation (box) and monosomy 7 are private to Ph53-D. Red and blue segments denote copy number gains and losses, respectively. **g**, *RUNX1* mutation is private to Ph53-R. *RUNX1* gene is on the minus strand and orange and blue represent G and C bases, respectively. A gray line represents a sequencing read, and a bar at the top represents the number of reads covering each position. Frequencies of reads with the c.445G>C mutation are shown. Lym, lymphoid progenitors; Mye, myeloid progenitors; GM, colony-forming unit (CFU)-granulocyte/macrophage; G, colony-forming unit-granulocyte; M, colony-forming unit-macrophage; E, burst-forming unit-erythroid; rel, relapse; D, diagnostic; R, relapse.

alterations in *IKZF1* are not responsible for the high-risk nature of *BCR-ABL1* lymphoblastic leukemia (Supplementary Figs. 20 and 21). Poorer outcomes of Inter-Pro patients are likely due to bi-allelic losses of *IKZF1* (ref. 38). Strikingly, the loss of *HBS1L*, exclusive to Early-Pro leukemias (Fig. 3a), was associated with extremely poor prognosis (Extended Data Fig. 9a). Early-Pro patients without the *HBS1L* deletion

showed similar survival outcome to Inter-Pro patients (Extended Data Fig. 9b). The role of *HBS1L* in the normal or malignant lymphoid cell is unknown. Interestingly, alterations specific to Early- and Inter-Pro sub-types showed similar prognostic value as the transcriptomic subtypes (Extended Data Fig. 9c), suggesting DNA- or RNA-based risk stratification may have clinical utility.

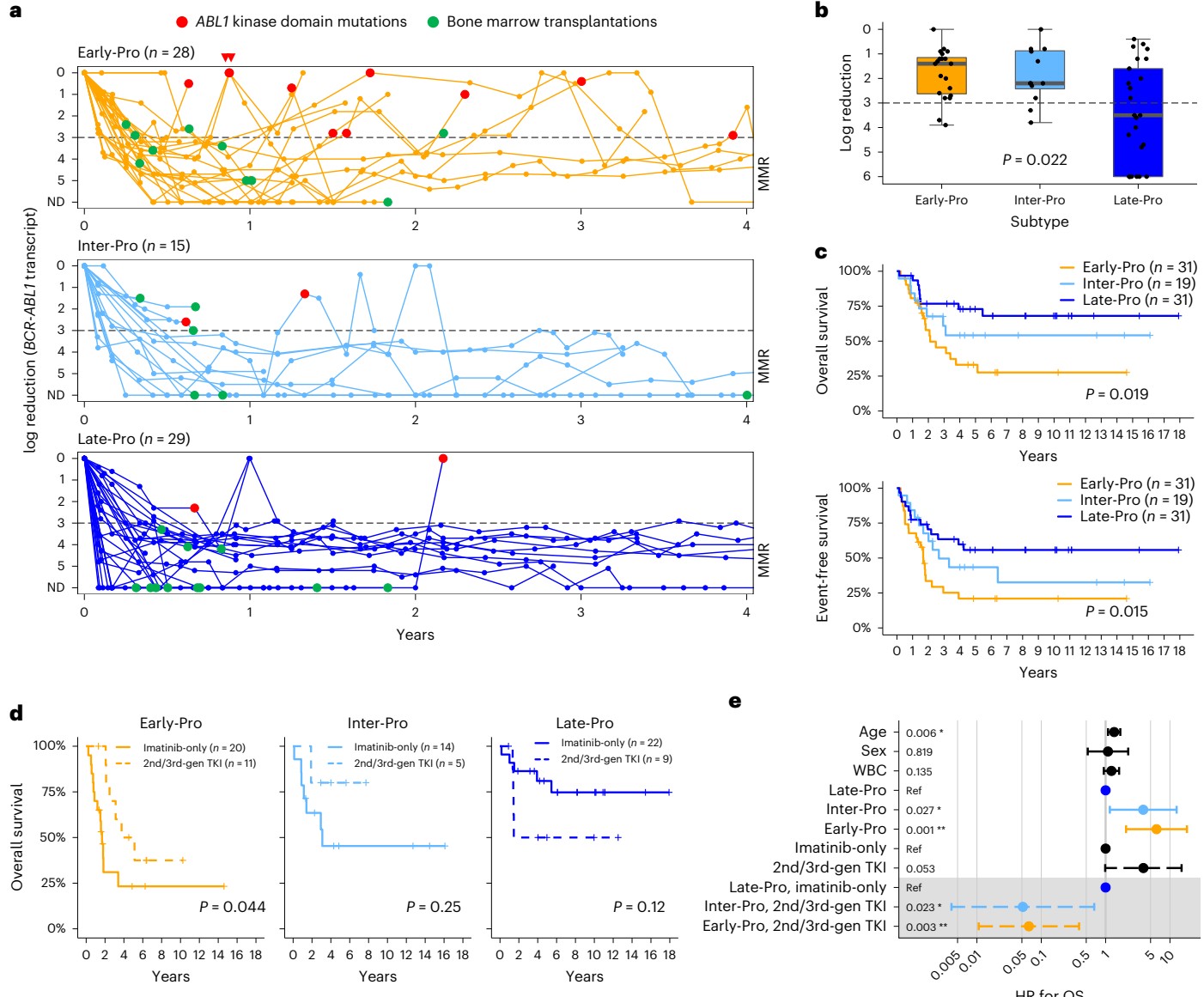

**Fig. 5 | Molecular subtypes predict treatment response and outcome.**
**a**, Residual disease plot of log reduction in *BCR-ABL1* transcript levels for each subtype. Each line represents a patient and each point represents a residual disease measurement. Dashed line denotes log reduction of 3, and log reduction of ≥3 is considered an MMR. Timings of first kinase domain mutation detections and BMTs are shown. **b**, First log reduction levels after induction therapy (at most 60 d from diagnosis) separated by subtype (20 Early-Pro, 12 Inter-Pro and 25 Late-Pro patients). Dashed line denotes log reduction of 3. *P* value from Kruskal–Wallis test is shown. **c**, Kaplan–Meier estimates of OS (top)

and EFS (bottom) by subtypes. **d**, Kaplan–Meier estimates of OS by treatment type (imatinib-only versus switch to second- or third-generation TKI) for each subtype. **e**, Multivariable Cox proportional hazards model incorporating age (scaled by 10), sex, WBC (scaled by median), subtype, treatment type and interaction of subtype and treatment type (gray shade; *n* = 81). Colors represent variables associated with Inter-Pro (light blue) and Early-Pro (orange). Numbers on the left are *P* values for each variable. ND, not detected; ref, reference. \*\**P* < 0.005 < \**P* < 0.05.

Next, we investigated TKI efficacy among the transcriptomic subtypes. Our cohort spanned diagnoses over two decades (1992–2019), during which multiple generations of TKIs against *BCR-ABL1* were approved, some of which are as follows: imatinib, dasatinib, nilotinib and ponatinib[39]. Imatinib, the first-generation TKI, targets the wildtype *BCR-ABL1*, whereas dasatinib, nilotinib and ponatinib, which are second- or third-generation TKIs, can target both wildtype and various mutant forms. Sixty-nine percent (*n* = 56/81) of patients received imatinib-only, and 31% (*n* = 25/81) were switched to second- or third-generation TKIs during their treatment. Early-Pro patients that received imatinib-only showed significantly poorer outcomes compared with those switched to a second- or third-generation TKI

(5-year OS: 23% versus 50%; *P* = 0.044, log-rank test; Fig. 5d). A similar trend was seen for Inter-Pro patients but was not conclusive due to a small sample size. Late-Pro patients showed a trend toward improved outcomes with imatinib-only therapies. Using the multivariable Cox proportional hazards model with interaction, Inter-Pro and Early-Pro subtypes showed a significant decrease in the hazard ratio (HR) when treated with second- or third-generation TKIs (Fig. 5e; Inter-Pro: HR = 0.05 (95% CI = 0.004–0.67), *P* = 0.023 and Early-Pro: HR = 0.07 (95% CI = 0.011–0.39), *P* = 0.003). This suggests that Early-Pro and Inter-Pro patients may specifically benefit from newer-generation TKIs, although this work needs further validation. Overall, the transcriptomic subtypes display differential sensitivity to TKIs.

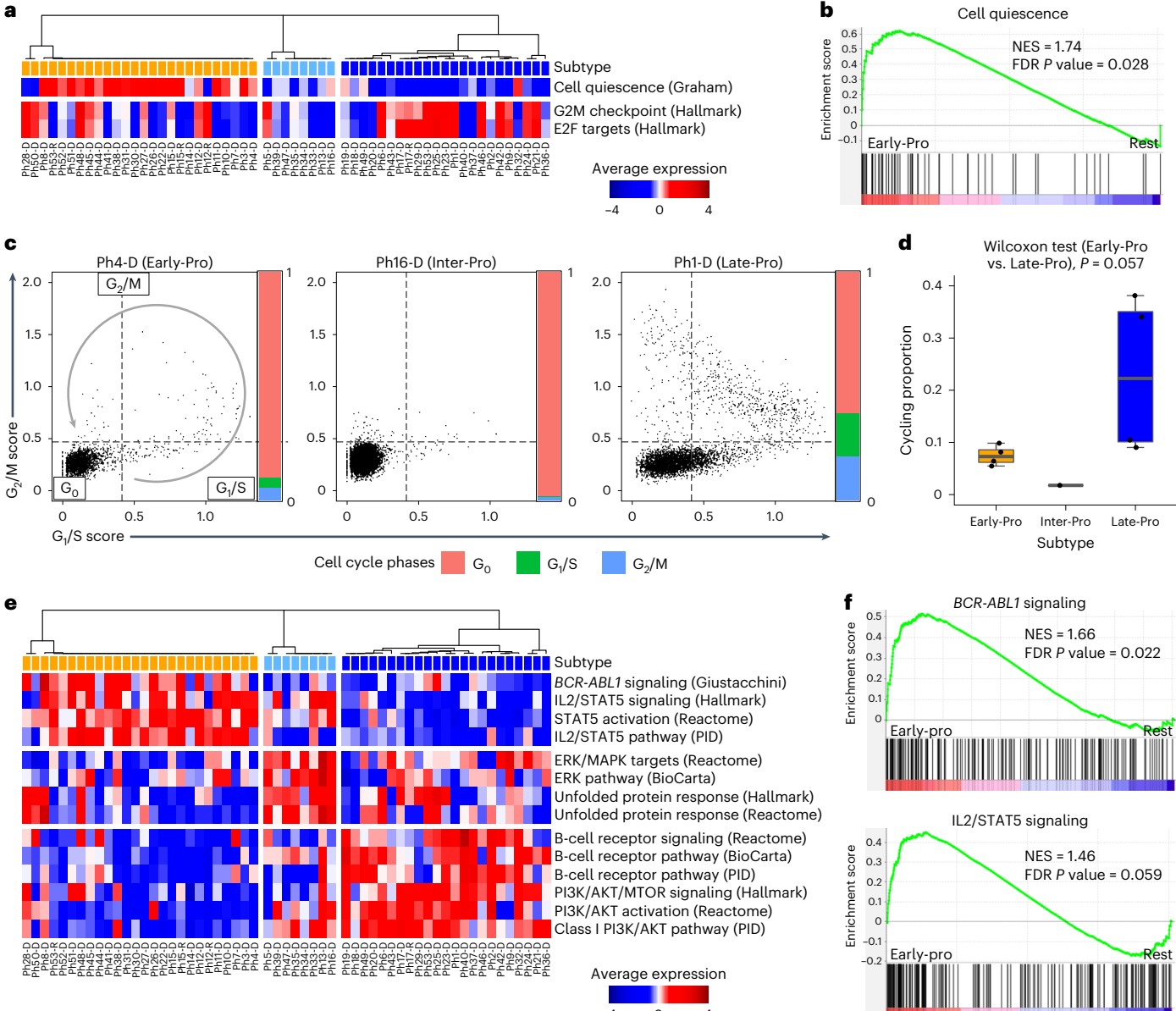

**Fig. 6 | Kinase domain-independent mechanisms of treatment resistance.**
**a**, Heatmap of average gene expression for cell quiescence and cell cycling gene sets. Sample order and subtype colors are the same as in Fig. 1a. **b**, Enrichment of cell quiescence gene set in the Early-Pro subtype (versus rest). **c**, Average $\log_2$ expression (score) of $G_1/S$ and $G_2/M$ phase gene sets in single cells. Exemplar cases from three subtypes are shown. Dashed lines denote the $G_1/S$ and $G_2/M$ cutoffs (one standard deviation greater than mean). Bars represent proportions of cells

in $G_0$, $G_1/S$ and $G_2/M$ phases. **d**, Proportion of cycling cells (non-$G_0$ phases) in each sample by subtype (four Early-Pro, one Inter-Pro and four Late-Pro leukemias). $P$ value from the Wilcoxon rank-sum test comparing Early-Pro and Late-Pro samples is shown. **e**, Heatmap of average gene expression for various signaling gene sets. Sample order and subtype colors are the same as in Fig. 1a. **f**, Enrichment of *BCR-ABL1* signaling and IL2/STAT5 signaling gene sets in the Early-Pro subtype (versus rest).

**Innate therapy resistance in Early-Pro and Inter-Pro leukemias**

Kinase domain mutations contributed to treatment resistance and poor survival in our cohort, particularly in those with Early-Pro leukemias[40] (Fig. 5a and Supplementary Fig. 22a). Interestingly, Early-Pro and Inter-Pro patients still showed worse survival compared with Late-Pro patients even after patients with kinase domain mutations were removed (Supplementary Fig. 22b). This suggested that these leukemias harbor kinase domain-independent mechanisms of TKI resistance. As noted earlier, Late-Pro leukemias were enriched for cell cycle-related programs. By contrast, the cell quiescence gene set was significantly enriched in Early-Pro leukemias[41,42] (Fig. 6a,b). In scRNA-seq data, Late-Pro leukemias contained more cells in $G_1/S$ and

$G_2/M$ phases (median 23%) than Early-Pro (8%) and Inter-Pro (2%) leukemias ($P = 0.057$, Wilcoxon rank-sum test for Early-Pro versus Late-Pro; Fig. 6c,d and Supplementary Fig. 23). Increased cell cycling in Late-Pro patients helps to explain their robust responses to induction therapy. Moreover, effective reduction of the leukemic load during induction likely reduces the opportunity for kinase domain mutations to develop.

In addition to cell quiescence, Early-Pro leukemias were also enriched for *BCR-ABL1* signaling and its downstream target STAT5 (Fig. 6e,f)[42–44]. Proteomics showed increased expressions of STAT5A and STAT5B, and immunodetection confirmed higher phosphorylation of STAT5 in Early-Pro samples (Supplementary Fig. 24a–c). By contrast, Inter-Pro leukemias showed marked upregulation of the UPR and ERK

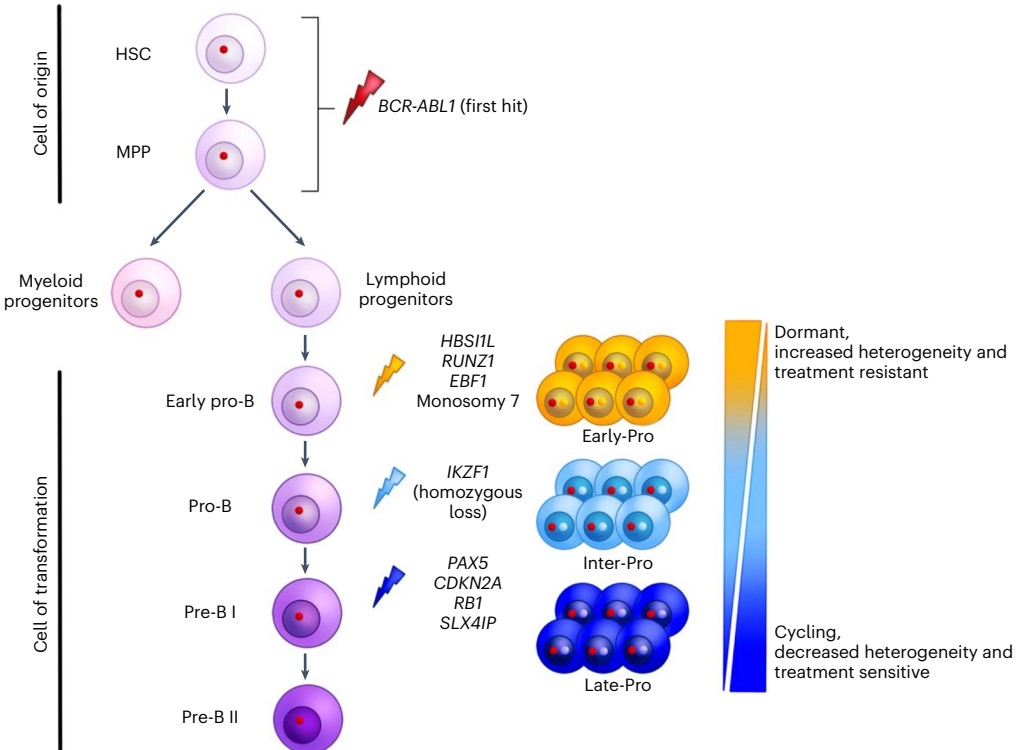

**Fig. 7 | Cell-of-transformation model for *BCR-ABL1* lymphoblastic leukemia subtypes.** The disease originates from the rearrangement of *BCR-ABL1* in a multipotent stem cell (HSC or MPP), which is the cell-of-origin. *BCR-ABL1* increases cell proliferation and impairs DNA damage response. It then transforms by acquiring cooperating alterations in a downstream B-cell progenitor, which is the cell-of-transformation. Transformation-related events are mostly generated by RAG-mediated recombination. Inactivation of B-cell transcription factors and tumor suppressors causes a block in differentiation and evasion of programmed cell death, respectively. The molecular subtypes of *BCR-ABL1* lymphoblastic leukemia arise from different target cells of transformation.

signaling pathways (Fig. 6e and Supplementary Fig. 24d). Increased STAT5 and UPR signaling are known mechanisms of TKI resistance in the absence of kinase domain mutations[45,46]. The interplay between cell quiescence and these kinase domain-independent resistance mechanisms promotes leukemic cell survival, which in turn may foster the development of kinase domain mutations at relapse in Early-Pro and possibly Inter-Pro patients (Extended Data Fig. 10).

Four patients (three Early-Pro and one Inter-Pro) in our cohort were switched from imatinib to a second- or third-generation TKI, dasatinib or ponatinib, after 2–6 months due to poor responses at induction. Kinase domain mutations were absent in all four leukemias indicating that they were innately resistant to treatment. Notably, all four patients achieved MMR after switching to a second- or third-generation TKI (Supplementary Fig. 25). This suggests that primary refractoriness in Early-Pro and Inter-Pro patients can be addressed with upfront use of more potent TKIs.

## Discussion

Resolving the transcriptomic subtypes improves our understanding of treatment response heterogeneity in *BCR-ABL1* lymphoblastic leukemia. One highlight of our work is that the emergence of therapy resistance is not random but is linked with the transcriptomic subtype of the disease. Identifying Early-Pro and Inter-Pro patients at diagnosis could impact treatment plans for these high-risk patients. As these patients often do not achieve DMRs by standard chemotherapy and imatinib, earlier transition to second- or third-generation TKIs before relapse may result in better responses. Clinically, our work also suggests that kinase domain-independent mechanisms of TKI resistance contribute to the development of kinase domain mutations.

From a tumor evolution perspective, our data have implications for how the transcriptomic subtypes of this disease develop.

Building on previous works[6,31,47], we propose a model of pathogenesis (Fig. 7). All transcriptomic subtypes are predicted to initiate from a common cell-of-origin that establishes a preleukemic phase. Isolation of preleukemic *BCR-ABL1*[+] clones capable of myeloid cell differentiation but lacking cooperating events suggests that the disease cell-of-origin, at least in some cases, is a multipotent cell. *BCR-ABL1* alone cannot drive transformation, and as preleukemic precursors enter lymphoid differentiation, the V(D)J recombination machinery is hijacked to drive the acquisition of cooperating events. Each step in normal B-cell differentiation (for example, early pro-B, pro-B or pre-B) is highly regulated by a unique set of transcription factors and presents a different molecular hurdle for transformation. Thus, the selection pressure changes during differentiation and shapes the genetic route of transformation. This model explains why each subtype is defined by cooperating events in genes that regulate a specific step of B-cell development. We posit that the differentiation stage at which the leukemia transforms, that is the 'cell-of-transformation' and not the cell-of-origin, determines the transcriptomic phenotype of the disease. Overall, our study provides a molecular classification scheme for *BCR-ABL1* lymphoblastic leukemia and provides important insights into the origins of this disease. It is possible that molecular programs that define each transcriptomic subtype can be targeted in combination with TKIs to improve the outcomes of patients with this aggressive disease.

## Online content

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

## Methods

### Patient samples

Diagnosis (*n* = 53) and relapse (*n* = 4) *BCR-ABL1* lymphoblastic leukemia samples were obtained from the University Health Network Leukemia Biobank (Supplementary Table 1). The cohort consisted of 44 B-ALL, 4 B/myeloid MPAL and 5 CML in lymphoid blast crisis (CML-LBC). The majority were adults except for Ph53, who was 13 years old at diagnosis and 18 years old at relapse (median age at diagnosis, 46 years; range, 13–83 years). Forty-nine samples were from peripheral blood (PB) and 8 samples were from BM biopsies. Samples were not selected based on any clinical criteria other than the detection of the Philadelphia chromosome and/or *BCR-ABL1* transcript. First, 27 patients were selected to have equal proportions of p190 and p210 isoforms, and the remaining patients were randomly selected. The cohort was composed of 30 p190 and 23 p210 *BCR-ABL1* isoforms. The study was granted approval by the Research Ethics Board of the University Health Network (REB 01-0573), and written informed consent was obtained from all patients.

### Statistical tests

All statistical tests were conducted in R v3.6.1 (ref. 50). In each boxplot, the middle line represents the median, lower and upper edges of the box represent the first and third quartiles, end of the lower whisker represents the smallest value at most 1.5× interquartile range from the lower edge of the box and the end of the upper whisker represents the largest value at most 1.5× interquartile range from the upper edge of the box. All data points were shown on the boxplots. In a few instances, a small number of extreme outliers were not shown, and their counts were shown in brackets. Fisher's exact test, Wilcoxon rank-sum test, and Kolmogorov–Smirnov test were performed as two-sided tests. Multiple comparison adjustments with the FDR method were made when applicable and noted as such. OS was computed as time from diagnosis to death or last follow-up, and EFS was computed as time from diagnosis to an event (either relapse or death) or last follow-up. Patients were right-censored at the last follow-up date.

### Fluorescence-activated cell sorting (FACS)

FACS was conducted to sort and analyze 57 *BCR-ABL1* ALL samples in the main cohort. The sorting scheme employed four fluorescently labeled, monoclonal, mouse antibodies, which are as follows: PE anti-CD19 (4G7; BD Biosciences, 349209; 4 µl per $10^6$ cells), APC-eFluor 780 anti-CD34 (4H11; eBioscience, 47-0349-42; 4 µl per $10^6$ cells), eFluor 450 anti-CD45 (2D1; eBioscience, 48-9459-42; 5 µl per $10^6$ cells) and Super Bright 645 anti-CD3 (OKT3; eBioscience, 64-0037-42; 3.5 µl per $10^6$ cells). B-cell lymphoblasts (100,000–200,000 cells) were collected and used for extraction of leukemia DNA and RNA, and polyclonal T-cells (30,000–50,000 cells) were collected and used for extraction of germline DNA (Extended Data Fig. 1b). From the first 30 samples (27 diagnoses and 3 relapses), variable numbers of HSC, MPP, MLP, CMP, GMP, MEP and mature B cells were also collected when possible[51]. This FACS scheme used the following eight additional antibodies: PerCP-eFluor710 anti-CD34 (4H11; eBioscience, 46-0349-42; 4 µl per $10^6$ cells), PE-Cy7 anti-CD38 (HIT2; BD Biosciences, 560677; 3.5 µl per $10^6$ cells), APC anti-CD90 (5E10; BD Biosciences, 559869; 5 µl per $10^6$ cells), FITC anti-CD45RA (HI100; BD Biosciences, 555488; 5 µl per $10^6$ cells), PE-Cy5 anti-CD49f (GoH3; BD Biosciences, 551129; rat; 4.5 µl per $10^6$ cells), AlexaFluor700 anti-CD10 (CB-CALLA; eBioscience, 56-0106-42; 5 µl per $10^6$ cells), biotin anti-FLT3 (4G8; custom conjugation from BD Biosciences; 8 µl per $10^6$ cells) and Qdot605 streptavidin (Thermo Fisher Scientific, Q10101MP; 3 µl per $10^6$ cells) for secondary staining of biotin anti-FLT3. Genomic DNA from rare cell populations (<2,000 cells) was amplified using the REPLI-g Mini Kit (Qiagen).

### RNA-seq

Total RNA was isolated using the PicoPure RNA Isolation Kit (Life Technologies) and DNA removal was performed using the RNase-free DNase

Set (Qiagen). Extracted RNA quantity was measured by Qubit RNA High Sensitivity Assay Kit (Thermo Fisher Scientific) and quality was assessed by running the samples on RNA ScreenTape Assay on the 2200 TapeStation System (Agilent Technologies) or RNA 6000 Nano Kit on the 2100 Bioanalyzer (Agilent Technologies). Only RNA with sufficient RNA integrity numbers (RIN; at least seven and mostly greater than eight) were used to generate libraries. cDNA sequencing libraries were prepared using the TruSeq RNA Access Library Prep Kit (Illumina) according to the protocols provided by the manufacturer. Using the KAPA Illumina Library Quantification Kits, library pools were quantified on the Eco Real-Time PCR Instrument. All libraries were sequenced on the Illumina HiSeq 2500 platform using paired-end cluster generation and 2 × 126 cycles.

### Gene expression quantification

RNA-seq data were aligned using STAR v2.5.2a (ref. 52) against human genome build 19 (hg19) and transcript sequences from Ensembl build 87. Default settings were used except for the following listed parameters: --chimSegmentMin 12 --chimJunctionOverhangMin 12 --alignSJDBoverhangMin 10 --alignMatesGapMax 100000 --alignIntronMax 100000 --chimSegmentReadGapMax parameter 3 --alignSJstitchMismatchNmax 5 -1 5 5. Gene counts were obtained from HTSeq v0.9.1 (ref. 53). Finally, the data were normalized using DEseq2 v1.18.1 (ref. 54) size factors and variance stabilizing transformation. Batch correction was performed using ComBat from the sva package[55,56].

### NMF and consensus hierarchical clustering

NMF was used to resolve expression signatures. NMF was performed on the top 2,000 variable genes using the NMF R package[57]. A total of 200 replicates were run for each *k* value between 2 and 10. Based on cophenetic scores, three NMF components with a total of 163 genes were identified. Subsequent consensus clustering was performed on these 163 genes utilizing ConsensusClusterPlus v1.40.0 (ref. 58). Using the Pearson correlation distance, clustering was performed using hierarchical clustering and complete linkage with 5,000 resamplings of 80% of samples per run to give rise to the final consensus. Based on the Silhouette scores, the relative change in cumulative distribution function between *k* values and the cophenetic coefficient, three main gene expression clusters were identified.

### Differential gene expression

Differential gene expression analysis was performed with DESeq2 v1.18.1 (ref. 54) using recommended settings. HTSeq counts were read into R, followed by size factor estimation, dispersion estimate and a negative binomial GLM fitting as implemented in DESeq2. Sex and batch were corrected in the linear mode. Differential expression was run across the three subtypes in a pairwise fashion. Next, genes were ranked based on the *P* value and the sign of the $\log_2$ fold change. Gene set enrichment was then conducted using GSEA Prerank v4.1.0 with recommended settings[59]. GSEA was run against the 50 Hallmark gene sets from MSigDB v7.2 (ref. 42) supplemented with the following four additional gene sets: *BCR-ABL1* signaling (genes upregulated in *BCR-ABL1*+ LSC versus normal HSC)[44], cell quiescence (genes upregulated in normal quiescent versus dividing CD34+ cells)[60], ERK signaling pathway (Reactome and Biocarta gene sets from the C2 curated list in MSigDB v7.2) and B-cell receptor pathway (Reactome and Biocarta gene sets from the C2 curated list in MSigDB v7.2).

### Gene fusion and antigen receptor loci rearrangement analyses

Gene fusions were identified using STAR-Fusion v1.0.0 (ref. 61) with default parameters. Antigen receptor loci rearrangements were called from RNA-seq fastq files using MiXCR v3.0.3 (ref. 62) with default parameters. Each leukemia was assessed for the expression of productive antigen receptor genes and V, (D) and J segment usages at the

following six loci: *IGH@*, *IGK@*, *IGL@*, *TRB@*, *TRA/D@* and *TRG@*. For the *IGH@*, which is rearranged in all samples in our cohort, SV breakpoint positions were also cataloged.

### Detection of residual disease

Residual disease levels of patients were assessed by a real-time quantitative PCR using a TaqMan primer-probe set for the *BCR-ABL1* fusion. The test was adapted from a published method[63,64] and performed by the Genome Diagnostics Laboratory at the University Health Network. Residual disease level was calculated as log reduction (base 10) relative to baseline level at diagnosis. The lower limit of detection was 5.0–5.5 log reduction.

### Reference dataset of hematopoietic cell compartments from human cord blood

Human cord blood samples with informed consent were obtained from Trillium Health, Credit Valley Hospital and William Osler Health System using procedures approved by the University Health Network Research Ethics Board. Freshly sorted populations from three to five independent cord blood pools (Supplementary Fig. 9a) were directly resuspended and frozen in PicoPure RNA Isolation Kit Extraction Buffer. RNA was extracted using PicoPure RNA Isolation Kit (Thermo Fisher Scientific) according to the manufacturer's protocols. RNA was analyzed on a Bioanalyzer Pico chip (Agilent Technologies), and samples that passed quality control according to integrity (RIN > 8) and concentration were subjected to further processing. Using SMART-Seq v4 Ultra Low Input RNA Kit for Sequencing (Takara), cDNA conversion was performed. Next, RNA-seq libraries were prepared using Nextera XT DNA Library Preparation Kit (Illumina) and equimolar quantities of libraries were pooled. Libraries were sequenced on the Illumina HiSeq 2500 for generating paired-end reads of 125 bp in length and a target sequencing depth of 55–75 million reads per sample.

### scRNA-seq and phenotype annotation

scRNA-seq was performed for nine *BCR-ABL1* ALL samples. Seven of nine samples were flow-sorted to enrich for leukemic blasts and two samples (Ph2-D and Ph22-D) were not flow-sorted. Using Chromium Single Cell 3′ v2 kit (10× Genomics), 10,000 leukemic cells from each sample were loaded to partition each cell and reagents into a droplet emulsion. scRNA-seq libraries were then prepared using the Single Cell 3′ v2 reagent kit (10× Genomics) according to the manufacturer's protocols. Initial library quantification was done using the Qubit dsDNA High Sensitivity Kit (Thermo Fisher Scientific). To assess library quality, samples were run on the LabChip GXII Touch HT (PerkinElmer) using the DNA High Sensitivity Kit (PerkinElmer). Sequencing libraries were quantified using the KAPA Illumina Library Quantification Kit (Roche) according to the manufacturer's instructions. Cluster generation and sequencing were performed on the Illumina HiSeq 2500 and NovaSeq 6000 platforms. For each sample, at least 40,000 mean reads per cell were acquired.

Using CellRanger v3.0.1 (10× Genomics), scRNA-seq data were aligned to hg19. This was followed by further processing using Seurat v3.1.2 (ref. 65). Cells with high ribosomal expression (>60%), high mitochondrial expression (>30%) or fewer than 1,000 expressed genes were removed from further analysis. A loess curve was fit to the UMI by gene curve, and cells that were found greater than two standard deviations away were removed. Furthermore, genes expressed in fewer than five cells were removed, along with mitochondrial and ribosomal genes. The data were normalized using the sctransform normalization method provided by Seurat and UMI count, percent mitochondria and cell cycle scores were regressed out. Normalized data were clustered using the self-assembling manifolds algorithm[66] with the Leiden clustering method at a resolution of 1. Uniform manifold approximation and projection (UMAP) were then performed[67]. To enrich each sample for leukemic cells, we removed cells that do not express *DNTT* but express *LYZ* (mono/gran), *GZMH* (NK) or *IL32* (NK/T) and cells that express

*CD3E*, *CD3G* or *CD8A* (T). When >60% of cells in a cluster showed those expression patterns, the whole cluster was removed.

Single cells from each leukemia sample were annotated with normal hematopoietic cell types using SingleR v1.0.6 (ref. 16). scTransform-normalized expression profile of each leukemia cell was iteratively compared to the reference dataset of bulk hematopoietic cell compartments (Kim et al., this study) to identify the cell type with the highest gene expression similarity. Pseudotime analysis was performed using Monocle 3 v0.2.3.0 (ref. 68). Based on the expression of *RUNX2*, which is a defining marker gene of C1/Early-Pro subtype, the starting point of each trajectory was derived. Cell cycle scores for each of the G1/S and G2/M phases were calculated using UMI counts normalized by library size (total UMI count) where cell cycle scores had not been regressed out. We used G1/S and G2/M genes from ref. 69, excluding M/G1 genes. Cells were scored for the G1/S and G2/M gene sets by taking the average $\log_2$ expression of the genes[70]. Each cell was then assigned to G0, G1/S or G2/M phase using cutoffs, which were determined for each gene set as one standard deviation greater than the mean score across all samples in our dataset. Cells below both the G1/S and G2/M cutoffs were assigned G0, cells exceeding the G1/S cutoff but below the G2/M cutoff were assigned G1/S, while remaining cells exceeding the G2/M cutoff were assigned G2/M.

To validate our findings, an independent dataset of adult BM cells from ref. 19 was also used. They profiled 13,165 adult BM cells via single-cell whole transcriptome sequencing capturing 30,781 unique transcripts and Abseq capturing 105 surface markers through BD Rhapsody. Cells spanning the normal B-cell development were selected for further analyses, which are as follows: 'HSCs and MPPs', 'Lymphomyeloid prog', 'Pre-pro-B cells', 'Pre-B cells', 'Pro-B cells', 'Small pre-B cells', 'Immature B cells', 'Mature Naive B cells', 'Nonswitched memory B cells' and 'Class switched memory B cells'. These totaled 3,032 cells. scRNA-seq data were normalized through scTransform with 2,000 variable genes, and a UMAP dimensionality reduction was performed with the following parameters: 30 PCs, 50 neighbors and cosine distance metric. Leiden clustering with multilevel refinement was performed at a resolution of 0.5, and clusters were labeled based on prior annotations, RNA marker genes, as well as key protein-level surface markers, including CD133, CD34, CD33, Tim3, CD38, CD45RA, CD10, CD19, CD20, CD21 and IgD. Then, Symphony v0.1.1 was used to project single leukemic cells from the current study onto the normal B-cell development hierarchy[71]. Briefly, Symphony was used to build a reference based on the scTransform-normalized expression of 2,000 variable genes. scRNA-seq data from 9 *BCR-ABL1* ALL samples were normalized with scTransform and projected onto the B-cell development reference map, and each query cell was assigned a cell type label based on 30 nearest-neighbors within the reference map.

### 3′ RNA-seq analysis of a second cohort

We obtained a second cohort of 40 *BCR-ABL1* ALL samples from the University Health Network Leukemia Biobank (Supplementary Table 3). The median age at diagnosis was 54 years old (range: 24–88 years old). Thirty-nine samples were from PB and one sample was from BM. The composition of *BCR-ABL1* isoforms in the cohort was 26 p190 and 14 p210. Human CD19 MicroBeads (Miltenyi Biotec) were used to enrich for CD19+ cells from 10 million bulk cells. While CD19− cells would be depleted by this method, we predicted this to have minimal impact as most samples in the main cohort showed blasts with high to dim levels of CD19. Furthermore, we previously observed that when blast populations with different antigen expression levels are taken from a given sample, their RNA profiles are highly consistent. RNA was extracted from CD19+ cells using PicoPure RNA Isolation Kit (Life Technologies) with RNase-free DNase Set (Qiagen). Extracted RNA quantity was measured by Qubit RNA High Sensitivity Assay Kit (Thermo Fisher Scientific) and quality was assessed using RNA ScreenTape Assay on the 2200 TapeStation System (Agilent Technologies).

3' RNA-seq libraries were constructed according to the Smart-3SEQ method as previously described[72]. In brief, 10 ng of RNA was fragmented and reverse-transcribed. The resulting double-stranded cDNA libraries were subjected to 17 PCR cycles for library amplification and index addition. Completed libraries were quantified by Qubit DNA High Sensitivity Assay Kit (Thermo Fisher Scientific) and quality was assessed on the LabChip GXII Touch HT using the DNA High Sensitivity Kit (PerkinElmer). Libraries were pooled according to their DNA concentrations and sequenced on the Illumina NextSeq 500 instrument with a Mid Output Reagent Kit v2.5 and 10% PhiX spike-in (Illumina), reading 150 nucleotides for read 1 and 6 nucleotides for the P7 index read. RNA-seq analysis was performed as described above. Consensus hierarchical clustering was performed using 163 genes identified from the original cohort.

## Whole-genome sequencing

Genomic DNA was extracted from leukemic blasts and T-cells of 57 samples in the main cohort using the Gentra Puregene Cell Kit (Qiagen). Extracted DNA was quantified using Qubit dsDNA High Sensitivity Kit (Thermo Fisher Scientific), followed by paired-end library preparations using the Nextera DNA Sample Prep Kit (Illumina) done according to the manufacturer's instructions. Next, prepared libraries were quantified using KAPA Illumina Library Quantification Kits on the Illumina Eco Real-Time PCR Instrument according to the manufacturer's protocols. All libraries were sequenced on the Illumina HiSeq 2000/2500 platform for 2 × 101 cycles with TruSeq Cluster kit v3 or 2 × 126 cycles with HiSeq Cluster kit v4 (Illumina).

## WGS data alignment and variant calling

WGS data were aligned using bwa v0.7.12 against human genome build 19 (hg19)[73] and postprocessing was done using Picard v1.90 (http://broad-institute.github.io/picard). Using the Genome Analysis Tool Kit (GATK) v3.6.0, germline variants were called according to the 'GATK best practices' for the version[74]. Strelka v1.0.7 (ref. [75]) and MuTect v1.1.4 (ref. [76]) were used to call somatic variants by applying default settings. The final somatic SNVs were determined by taking the intersect of 'Tier 1' SNVs from Strelka and the 'PASS' variants from MuTect. Strelka was also used to identify somatic indels. Copy number variations (CNVs) were identified using HMMcopy v0.1.1 (ref. [77]). Somatic SVs were determined as the union of filtered calls from two tools, CREST alpha[78] and Delly v0.5.5 (ref. [79]). Variants were inspected using Integrated Genomics Viewer[80], and CNVs and SVs were annotated with overlapping genes and repetitive sequence elements to highlight biologically meaningful events, such as gene fusions, and filter out potential false positives.

## Nested PCR

SVs and SNVs were validated by a custom touchdown, nested PCR with leukemia-specific primers followed by Sanger sequencing. For SVs, such as translocations and deletions, two pairs of primers were designed to anneal to the opposite sides of the breakpoint junction (Supplementary Table 13). Presence or absence of PCR amplification corresponded to presence or absence of the variant. To enhance specificity, the internal primer pair for each variant was designed to only amplify the PCR amplicon of the external primer pair, and the size difference between the two amplicons was at least 100 bp. PCR was carried out using the Phusion High-Fidelity PCR Master Mix with HF Buffer (New England Biolabs), and custom DNA oligos were acquired from Integrated DNA Technologies. Sanger sequencing was performed at the Centre for Applied Genomics (TCAG) DNA Sequencing Facility at the Hospital for Sick Children, Toronto.

## SiMSen-seq for detection of low-frequency mutations in *ABL1* kinase domain

SiMSen-seq was performed according to the developer's protocol[81] to detect low-frequency mutations in the kinase domain of *ABL1*. In brief,

standard PCR primers of 18–27 nucleotides in length were designed using BatchPrimer3 (ref. [82]) to generate eight amplicons covering frequently mutated regions in the kinase domain (amino acids 221–266, 275–424, 428–473 and 475–504). Standard primers with the highest amplification efficiency and specificity were chosen for each amplicon, and barcoding primers were designed by adding unique molecular barcode and adapter sequences. Final amplicon sizes ranged from 281 to 300 bp. Barcoding primers were pooled to 1 µM each (primer pairs with relatively lower efficiency were pooled at slightly higher concentrations). 70 ng of genomic DNA from 48 samples (44 diagnostics and 4 relapses) were used for each SiMSen-seq reaction. Initial barcoding PCR (three cycles) and adapter amplification PCR (27 cycles) were performed on the C1000 Thermal Cycler (Bio-Rad). Libraries were run on the LabChip GXII Touch HT using the DNA High Sensitivity Kit (PerkinElmer) to assess the quality and quantified using Qubit dsDNA High Sensitivity Kit (Thermo Fisher Scientific). Paired-end sequencing (2 × 150) was performed on the Illumina NextSeq 500 instrument with a Mid Output Reagent Kit v2.5 and 15% PhiX spike-in (Illumina). Debarcer algorithm was used to align sequence data, build sequence consensus and call variants[81]. Reads that map to the same amplicon and have the same molecular barcode are termed family of reads, and those with minimum family size of 20 were used to build consensus reads and remove PCR errors. The threshold for detection of low-frequency mutations was either three or more supporting consensus reads with consensus mutation frequency >0.4% or five or more supporting consensus reads.

## Automated, capillary western blot assay

Proteins were extracted from cell pellets (1–2 million cells) by lysing with RIPA buffer (Thermo Fisher Scientific), 1× HALT Protease and Phosphatase Inhibitor Cocktail (Thermo Fisher Scientific) and 5 mM EDTA. Samples were pipette mixed and placed on ice for 5 min, followed by centrifugation at 15,000g at 4 °C for 15 min to pellet debris. The supernatant containing the extracted protein was transferred to a new tube. The protein concentration was determined using the Pierce BCA Protein Assay Kit (Thermo Fisher Scientific) according to the manufacturer's protocols.

Protein analyses were conducted on the western blot assay system (ProteinSimple) according to the manufacturer's protocols. The 12–230 kDa separation module was used for phosphorylated-STAT5 (C11C5; Cell Signaling Technology, 9359; 1:100 dilution) and total protein detection assays. In brief, extracted protein was diluted with 0.1× Sample Buffer and 5× Master Mix and heat-denatured for 5 min at 95 °C before loading. For the phosphorylated-STAT5 and c-Abl assays, the primary antibodies were diluted with Milk-Free Antibody Diluent and Antibody Diluent 2, respectively. All reagents were added to a microplate and a capillary cartridge was loaded on the western blot assay instrument, and run parameters were set using the COMPASS software v4.0.0 (ProteinSimple). The corresponding band of each protein was visualized using the COMPASS software, and protein concentration of the band was normalized to the total protein content.

## Mass spectrometry

Samples (100,000 flow-sorted blasts) were lysed by repeated freeze-thaw cycles (five cycles, switching between a dry ice/ethanol bath and 60 °C water bath) in lysis buffer containing 50 mM HEPES pH 8, 1% Triton X-100, 1% SDS, 1% Tween 20, 1% NP-40, 1% deoxycholate, 5 mM EDTA, 50 mM NaCl, 1% glycerol and 1% complete protease inhibitor (Roche)[83,84]. Samples were sonicated on a probe-less ultrasonic sonicator for five 10-s cycles at 10 watts per tube (Hielscher VialTweeter) to shear genomic DNA. This was followed by centrifugation at 18,500g to pellet cell debris. The supernatant was aspirated for subsequent steps.

SUC2 protein (yeast invertase) of 2 pmol was added as a digestion control. Disulfide bonds were reduced with 5 mM dithiothreitol, followed by a 30-min incubation at 60 °C. Free sulfhydryl groups were alkylated

by incubating samples in 25 mM iodoacetamide in the dark at room temperature for 30 min. An additional 5 mM of dithiothreitol (DTT) was added to quench the alkylation reaction and samples were incubated at room temperature for 5 min. The magnetic bead-based SP3 protocol[85] was used to capture proteins before digestion. In brief, magnetic beads were added to proteins in a 10:1 (wt/wt) ratio. Absolute ethanol was added to bring the ethanol concentration to 50%. Samples were shaken at room temperature for 5 min at 1,000 rpm, and supernatant was discarded. The beads were rinsed three times with 80% ethanol and air-dried. Proteins were digested in 25 mM ammonium bicarbonate containing 2 µg of trypsin/Lys-C enzyme mix (Promega) at 37 °C overnight. Samples were centrifuged at 18,500g for 1 min to pellet the beads, and the supernatant containing digested peptides was removed. Hundred microliter of water was used to elute remaining peptides off the beads. Peptides were desalted by C18-based solid phase extraction, then lyophilized in a SpeedVac vacuum concentrator. Peptides were solubilized in mass spectrometry-grade water with 0.1% formic acid.

Synthetic iRT peptides (Biognosis) were spiked into each sample at a 1:10 ratio before data acquisition. LC–MS/MS data were acquired using an Easy nLC 1000 (Thermo Fisher Scientific) nano-flow liquid chromatography system with a 50 cm EasySpray ES803 column (Thermo Fisher Scientific) that is coupled to Orbitrap Fusion tandem mass spectrometer (Thermo Fisher Scientific). Peptides were separated by reverse phase chromatography using a 4-h nonlinear chromatographic gradient of 4–48% buffer B (0.1% formic acid (FA) in acetonitrile (ACN)) using a flow rate of 250 nl min$^{-1}$. Column temperature was kept at 45 °C. Mass spectrometry data were acquired in XCalibur 4.2.47 in positive-ion data-dependent mode. MS1 data were acquired at a resolution of 120,000 in the orbitrap, AGC target of $1 \times 10^6$ and maximum injection time (maxIT) of 50 ms and 40 s dynamic exclusion, while MS2 data were acquired in the ion trap at 'Normal' scan rate, maxIT of 100 ms. Higher energy collisional dissociation (HCD) fragmentation was performed at a normalized collision energy of 31%. Data were searched in MaxQuant v1.6.3.3 (ref. [86]) using a merged UniProt protein sequence database containing human protein sequences from UniProt (complete human proteome, 2015-01-27; number of sequences 42,842), Suc2 (yeast) protein sequences from UniProt and iRT synthetic peptide sequences (Biognosis). Searches were performed with a maximum of two missed cleavages, and carbamidomethylation of cysteine as a fixed modification. Variable modifications were set as oxidation at methionine and acetylation (N-term). The false discovery rate for the target-decoy search was set to 1% for protein, peptide and site levels. Intensity-based absolute quantification (iBAQ), label-free quantitation (LFQ) and match between runs (matching and alignment time windows set as 0.7 min and 20 min, respectively) were enabled. The proteinGroups.txt file was used for subsequent analysis. Proteins matching decoy and contaminant sequences were removed, and proteins identified with two or more unique peptides were carried forward. LFQ intensities were used for protein quantitation[87]. For proteins with missing LFQ values, median-adjusted iBAQ values were used as replacement[88]. Protein intensities were log$_2$-transformed for further analysis (Supplementary Table 14).

## Analysis of SV breakpoints and junctions

SVs (deletions, insertions, inversions, intrachromosomal and interchromosomal translocations) were delineated to nucleotide-level resolution and analyzed for breakpoint and junction characteristics. Cryptic RSS were defined as the canonical heptamer ('CACAGTG'), its reverse complement ('CACTGTG') or variants retaining the first 'CACA' and a subset of other canonical bases at nonantigen receptor loci[28,89]. SVs were categorized as RSS$^+$ or RSS$^-$ based on the presence of RSS motifs on the inside of both breakpoints. When two breakpoints of an SV separate genomic regions into segments 1A/1B and 2A/2B, where segments 1A

and 2B are then joined together, search ranges for RSS motifs consisted of the first 30 bp of 1B and 2A. A few exceptions to the rule were made, which are as follows: (1) If an RSS motif is found at one breakpoint, the other RSS is allowed to be a variant retaining the first 'CAC' and a subset of other canonical bases; (2) If an RSS is found within 30 bp on the inside of one breakpoint, the other RSS is allowed to be located further from the breakpoint (>96% of RSS are found within 30 bp and >99% are found within 50 bp); (3) If a breakpoint is near a breakpoint cluster, such as those in *IKZF1*, the RSS is allowed to be located further from the breakpoint; (4) A less conserved RSS motif is considered functional if it is near a cluster of breakpoints. Notably, 'CATACTG' was found near a breakpoint cluster in *LEMD3*, and 'CCCAGTG' was found near a breakpoint cluster in *PCMTD1*. In total, 796 cryptic RSS motifs were identified from 399 RSS$^+$ cooperating event SVs; one breakpoint in *IKZF1* and one in *CBWD2* did not harbor RSS motifs despite the opposite breakpoints being positioned in RSS$^+$ breakpoint clusters. Sequence logos were generated using WebLogo v2.8.2 (ref. [48]).

H3K4me3 ChIP–seq signal of GM12878 (B-lymphoblast cell line) from the ENCODE database was used to calculate the distance between each SV breakpoint and the nearest H3K4me3 peak[90]. Chromatin state annotations of GM12878 by ChromHMM were used to assign each SV breakpoint with a chromatin state[29,91]. SV junctions were assessed for NTS insertion (addition of nucleotides not present at either breakpoint), microhomology (identical sequence of nucleotides between two breakpoints) or clean junction (neither NTS insertion nor microhomology)[47].

## Colony formation assay

CD34$^+$CD19$^-$CD45RA$^-$ cells from 14 diagnostic samples were collected by FACS using the following seven mouse antibodies: PE anti-CD19 (4G7; BD Biosciences, 349209; 4 µl per 10$^6$ cells), APC-eFluor 780 anti-CD34 (4H11; eBioscience, 47-0349-42; 4 µl per 10$^6$ cells), PerCP-eFluor710 anti-CD34 (4H11; eBioscience, 46-0349-42; 4 µl per 10$^6$ cells), FITC anti-CD45RA (HI100; BD Biosciences, 555488; 5 µl per 10$^6$ cells), PE-Cy7 anti-CD38 (HIT2; BD Biosciences, 560677; 3.5 µl per 10$^6$ cells), eFluor 450 anti-CD45 (2D1; eBioscience, 48-9459-42; 5 µl per 10$^6$ cells) and AlexaFluor700 anti-CD10 (CB-CALLA; eBioscience, 56-0106-42; 5 µl per 10$^6$ cells). For colony formation, the collected cells were mixed with 1 ml or 2 ml of MethoCult H4034 Optimum methylcellulose medium (STEMCELL Technologies, 04034) and plated onto one or two 35-mm Petri dishes. Individual colonies were phenotyped after 13 d as a colony-forming unit (CFU)-GM, CFU-G, CFU-M, or burst-forming unit-erythroid (E). Colonies were individually collected into 1.5 ml tubes, washed in PBS with 5% FBS, lysed for DNA using Arcturus PicoPure DNA Extraction Kit (Life Technologies; 15 µl per colony) and transferred to 96-well PCR plates. Patient- and variant-specific nested PCR was performed with 1 µl of each colony DNA.

## Reporting summary

Further information on research design is available in the Nature Portfolio Reporting Summary linked to this article.

## Data availability

WGS and RNA-seq data are deposited and publicly available on the European Genome-Phenome Archive (EGAS00001007167). Mass spectrometry data can be found in Supplementary Table 14.

## Code availability

We performed all analyses using publicly available software as described and referenced in Methods.

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

## Acknowledgements

We thank the patients and families for their consent and the Leukemia Tissue Bank at the University Health Network for providing leukemia samples. We also thank members of the Notta Laboratory for their critical review of the paper and the Ontario Institute for Cancer Research (OICR) Genomics Technology program, the SickKids-UHN Flow Cytometry Facility and the Princess Margaret Genomics Centre for their technical support. F.N. is supported by the Gattuso-Slaight Personalized Cancer Medicine Fund from the Princess Margaret Cancer Foundation and the OICR and is a recipient of the Ontario Early Researcher Award (ER19-15-205), the V Foundation Scholar Award (V2020-016) and the American Society of Hematology Scholar Award (1018787). J.D.M. was supported by funding from the OICR. M.D.M. was

supported by the Orsino Chair in Leukemia Research, the Canadian Institutes of Health Research (CIHR) and the Leukemia & Lymphoma Society of Canada. J.C.K. was supported by the Vanier Canada Graduate Scholarship from the CIHR (294582).

## Author contributions

J.C.K. designed and performed the experiments and contributed to data analyses. M.C.-S.-Y., S.G., A.G.X.Z., A.Z. and S.C. contributed to the bioinformatic analyses. K.N., O.I.G., L.G.-P., E.F.-F., S.M.D., A.K. and S.K. contributed to the experiments. T.W. performed model illustrations. A.A., N.I. and A.G. performed clinical annotation. A.T. provided clinical flow cytometry data. J.H. and M.D.M. provided clinical samples. T.K., J.E.D., J.D.M., M.D.M. and F.N. supervised the study. J.C.K. and F.N. wrote the paper.

## Competing interests

The authors declare no competing interests.

## Additional information

**Extended data** is available for this paper at https://doi.org/10.1038/s41588-023-01429-4.

**Correspondence and requests for materials** should be addressed to Faiyaz Notta.

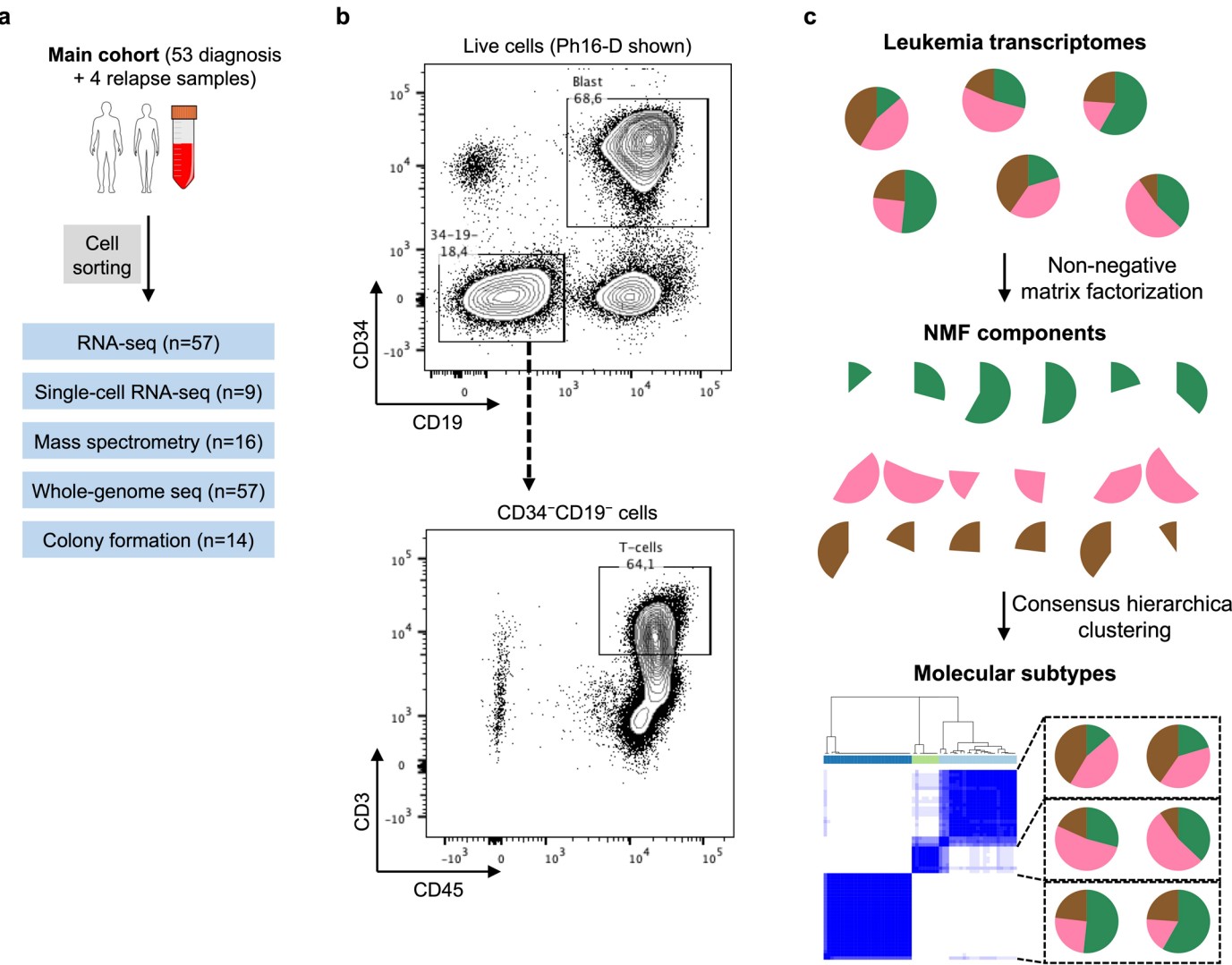

**Extended Data Fig. 1 | Flow sorting and sample clustering workflows. a**, Study design of the main cohort. **b**, Exemplar fluorescence-activated cell sorting scheme for leukemic blasts and T-cells. **c**, RNA-seq clustering approach incorporating non-negative matrix factorization (NMF) and consensus hierarchical clustering.

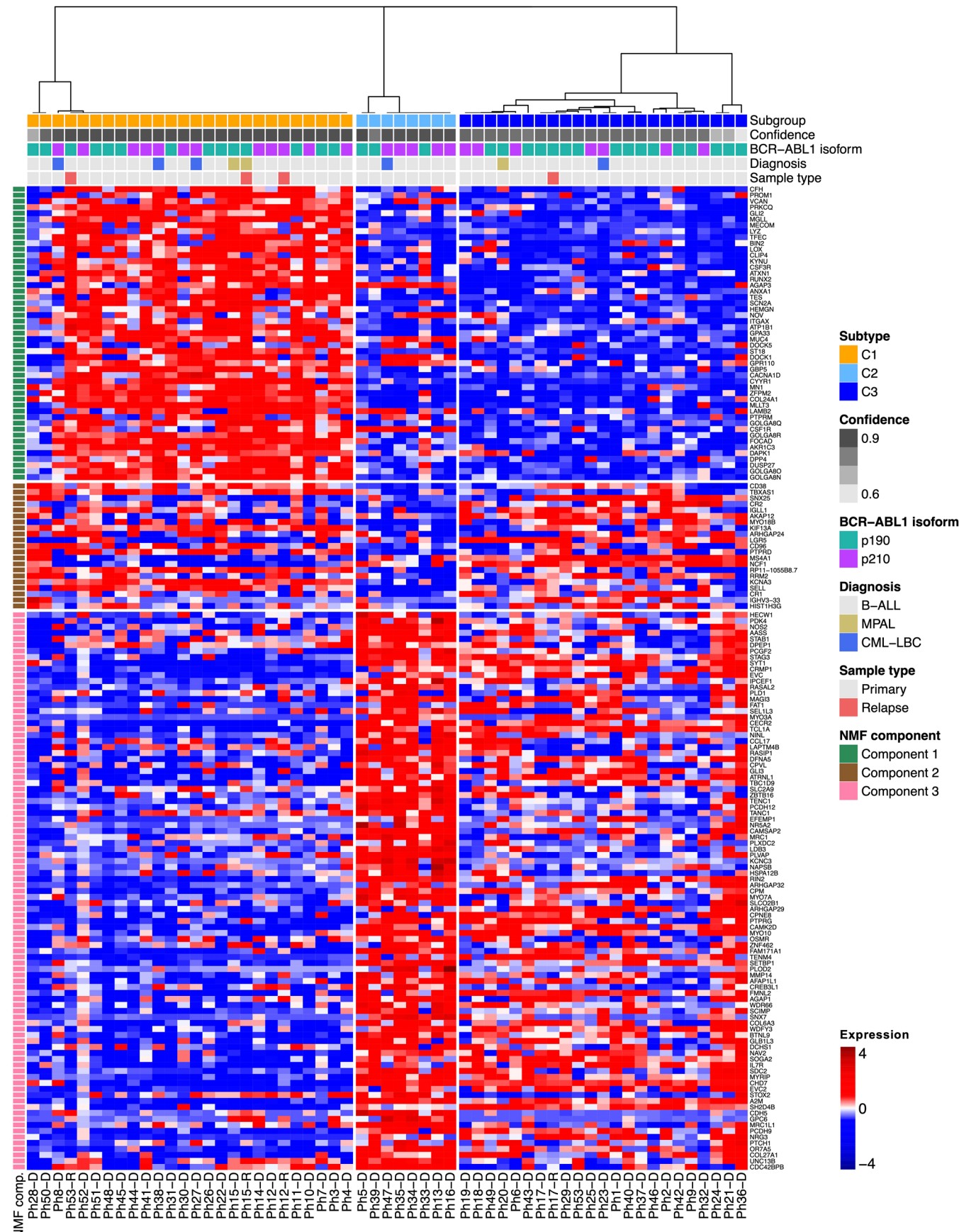

**Extended Data Fig. 2 | Three molecular subtypes of *BCR-ABL1* lymphoblastic leukemia.** Consensus hierarchical clustering using 163 NMF component genes produced three molecular subtypes of *BCR-ABL1* lymphoblastic leukemia. Rows represent the genes grouped into NMF components, and columns represent the 57 samples. Molecular subtypes, consensus clustering confidence scores, *BCR-ABL1* isoforms, diagnoses (B-ALL, MPAL, CML-LBC), and sample types (primary/diagnostic, relapse) are shown in tracks.

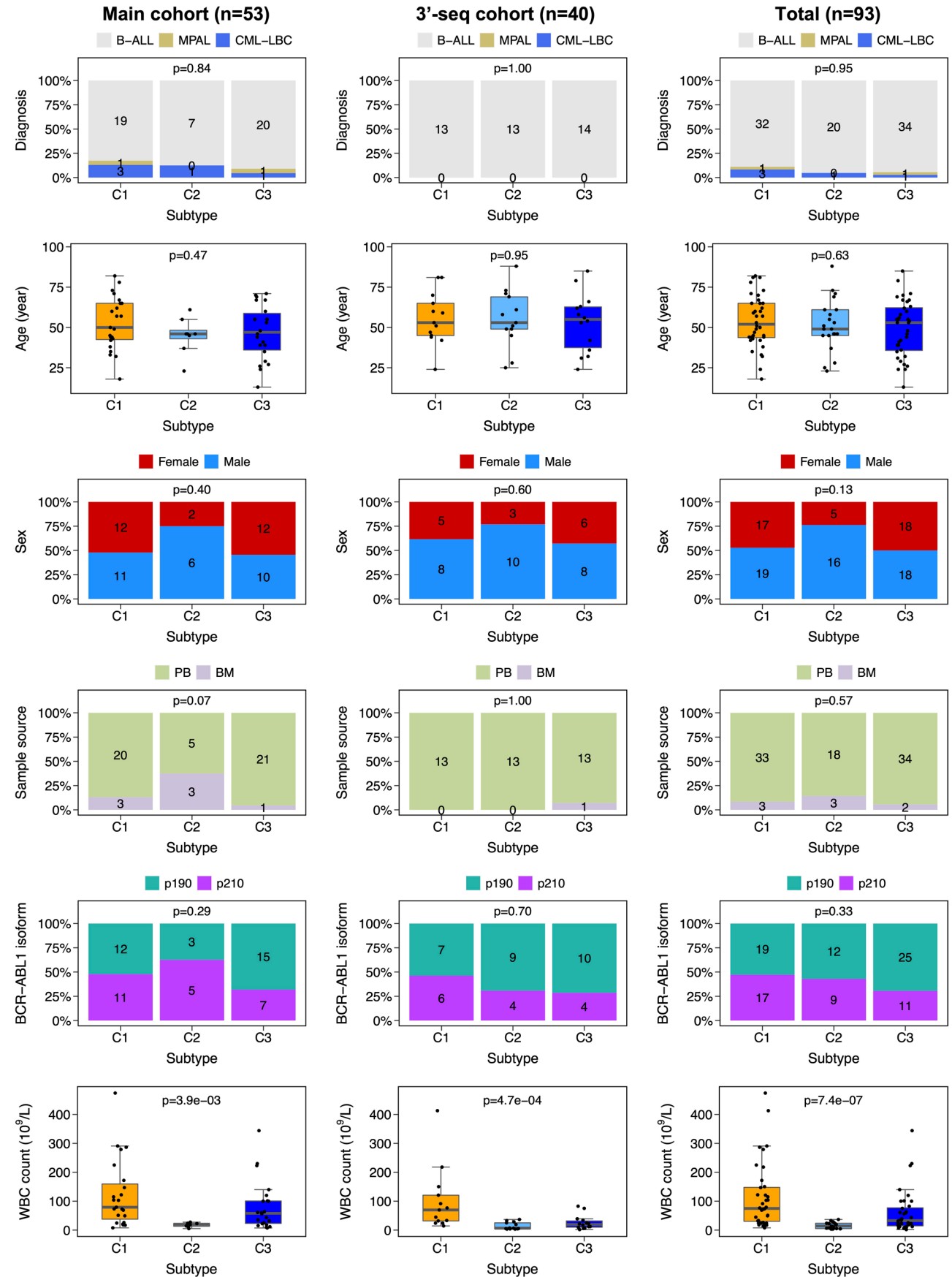

**Extended Data Fig. 3 | Patient and sample characteristics of three transcriptomic subtypes.** Comparison of clinical and sample characteristics of three molecular subtypes. Plots are separated into main cohort (left; 23 C1, 8 C2, 22 C3), 3'-seq cohort (centre; 13 C1, 13 C2, 14 C3), and combined cohort (right; 36 C1, 21 C2, 36 C3). Counts represent numbers of patients/primary leukemias. Kruskal-Wallis test was used for comparing age at diagnosis and WBC, and Fisher's exact test was used for the rest. PB, peripheral blood; BM, bone marrow; WBC, white blood cell.

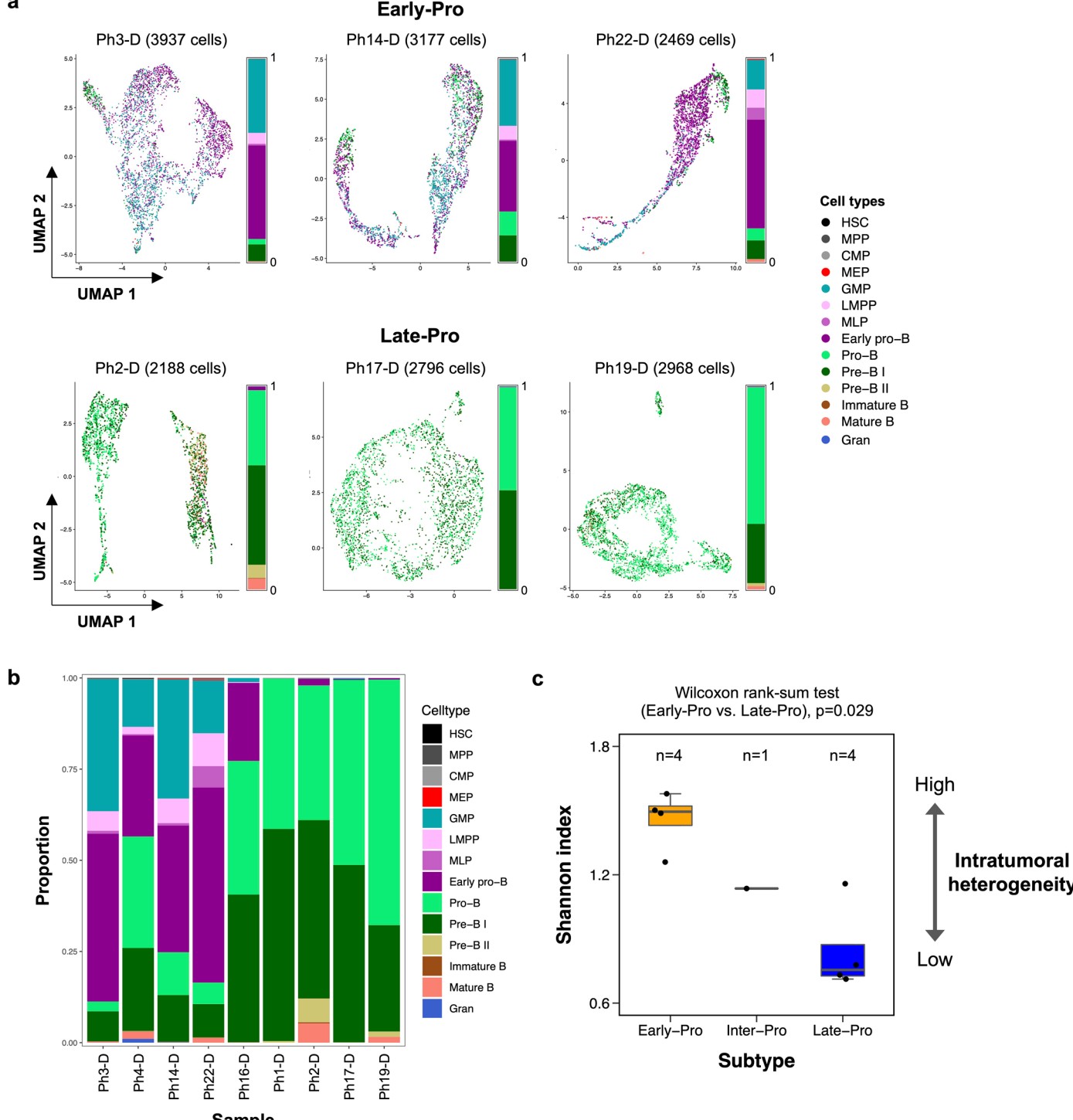

**Extended Data Fig. 4 | Cell type annotation of single-cell RNA-seq data.**
**a**, Single cells from scRNA-seq are annotated with their closest normal cell counterparts and visualized using UMAP. Six cases (3 Early-Pro and 3 Late-Pro) not shown in Fig. 2f are shown here. Bars represent proportions of annotated cell types. **b**, Proportions of annotated cell types in nine scRNA-seq samples. **c**, Shannon diversity index of cell type annotations in each single-cell RNA-seq sample. p-value from Wilcoxon rank-sum test comparing Early-Pro and Late-Pro subtypes is shown.

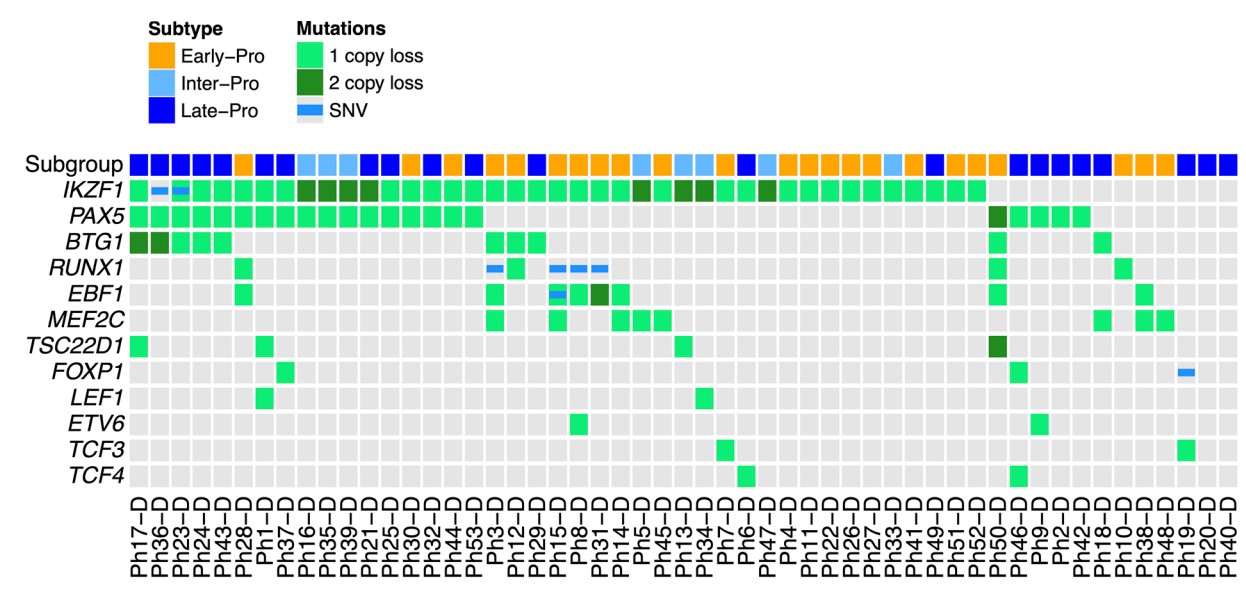

**Extended Data Fig. 5 | Frequent inactivation of lymphoid or B-cell development genes. a**, Oncoprint of alterations in lymphoid or B-cell development genes in decreasing order of recurrence rate. **b**, Histogram of the number of lymphoid or B-cell development genes altered in diagnostic leukemias. Mean number is 2.1. **c**, Frequency of homozygous losses of *IKZF1* by subtype. p-value from Fisher's exact test is shown. **d**, Frequency of dominant negative *Ik6* deletion by subtype. p-value from Fisher's exact test is shown.

**e**, Gene expression score for stromal cell signature[15] by subtype (26 Early-Pro, 8 Inter-Pro, 23 Late-Pro leukemias). **f**. Normalized counts (RNA-seq) of *IL7R* transcripts by subtype (26 Early-Pro, 8 Inter-Pro, 23 Late-Pro leukemias). **g**, Mutation loads (SNV and indel counts) by subtype. p-values from pairwise Wilcoxon rank-sum tests and Kruskal-Wallis test are shown. Data from primary leukemias with a median genome coverage of 15.8x are shown (12 Early-Pro, 3 Inter-Pro, 13 Late-Pro).

**a**

**Canonical RSS heptamer**

Frequency

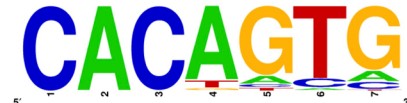

Information content

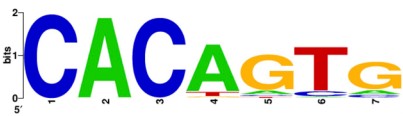

**b**

**Cryptic RSS heptamers from *BCR-ABL1* SVs with RSS motifs (n=20 breakpoints)**

Frequency

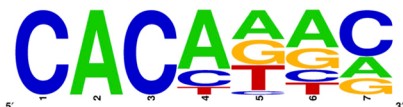

Information content

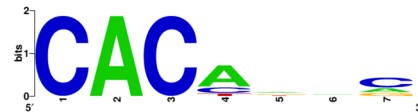

**c**

**Cryptic RSS heptamers from *BCR-ABL1* SVs with RSS motifs (n=20 breakpoints)**

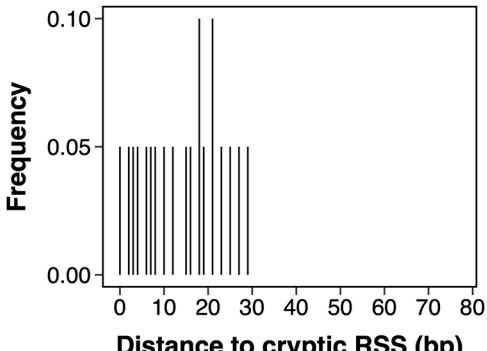

**d**

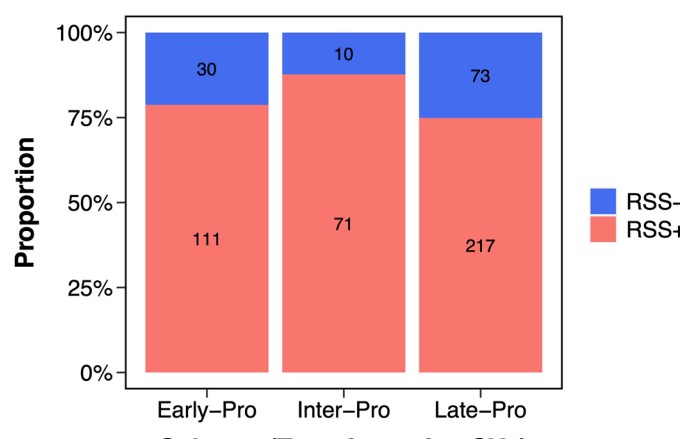

**Extended Data Fig. 6 | Detection and analysis of cryptic RSS sequences SV breakpoints. a**, Sequence logo of canonical RSS heptamers[28]. **b**, Sequence logo of cryptic RSS heptamers from *BCR-ABL1* SVs with RSS motifs (n = 20). Last 3 bases are not conserved for 'GTG'. **c**, Distances between SV breakpoints and nearest RSS motifs for *BCR-ABL1* SVs with RSS motifs. Unlike RSS motifs from transformation related SVs, they do not form a negative binomial distribution. **d**, Proportions of transformation SVs with and without RSS motifs by subtype.

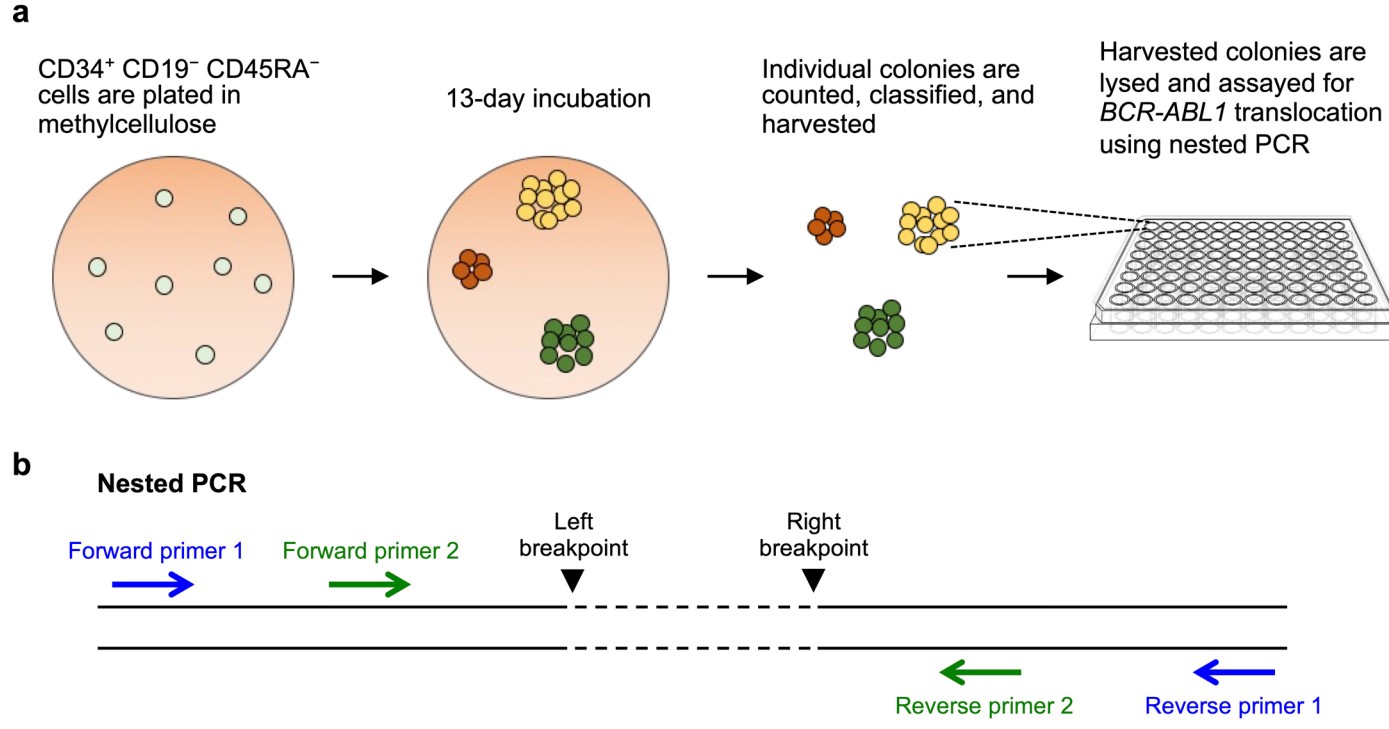

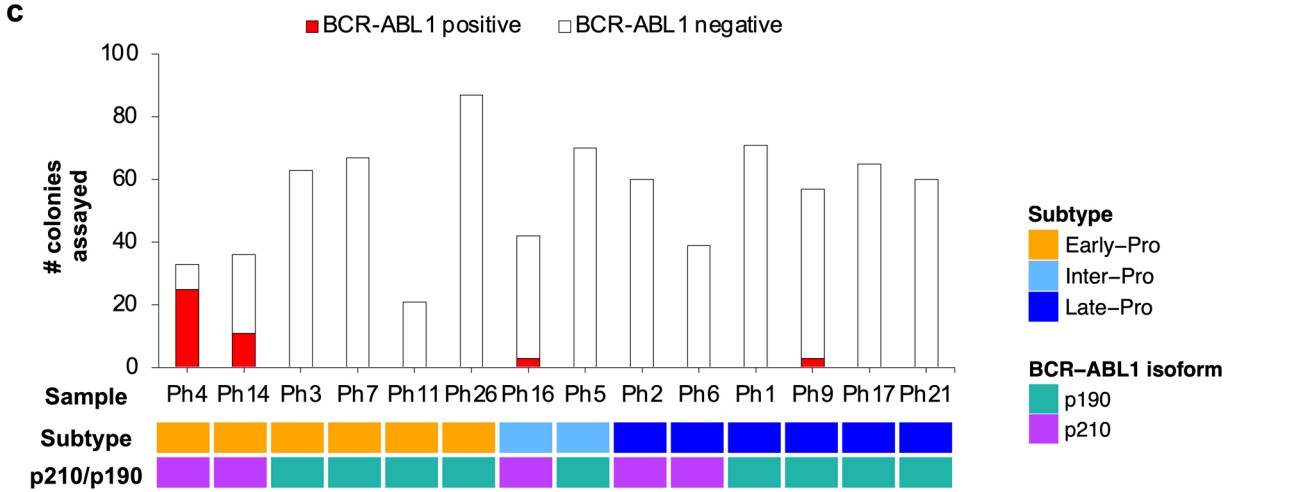

**Extended Data Fig. 7 | Colony formation assay. a**, Schematic representation of the colony formation assay. **b**, Schematic representation of the nested PCR. Two breakpoints of the assayed SV are located >5 kb apart on the same chromosome or on different chromosomes (for example *BCR* in chr22 and *ABL1* in chr9). PCR amplification is successful only when the assayed SV is present. **c**, Numbers of *BCR-ABL1* positive and negative colonies in 14 samples. Subtype and *BCR-ABL1* isoform (p210/p190) of each sample are shown.

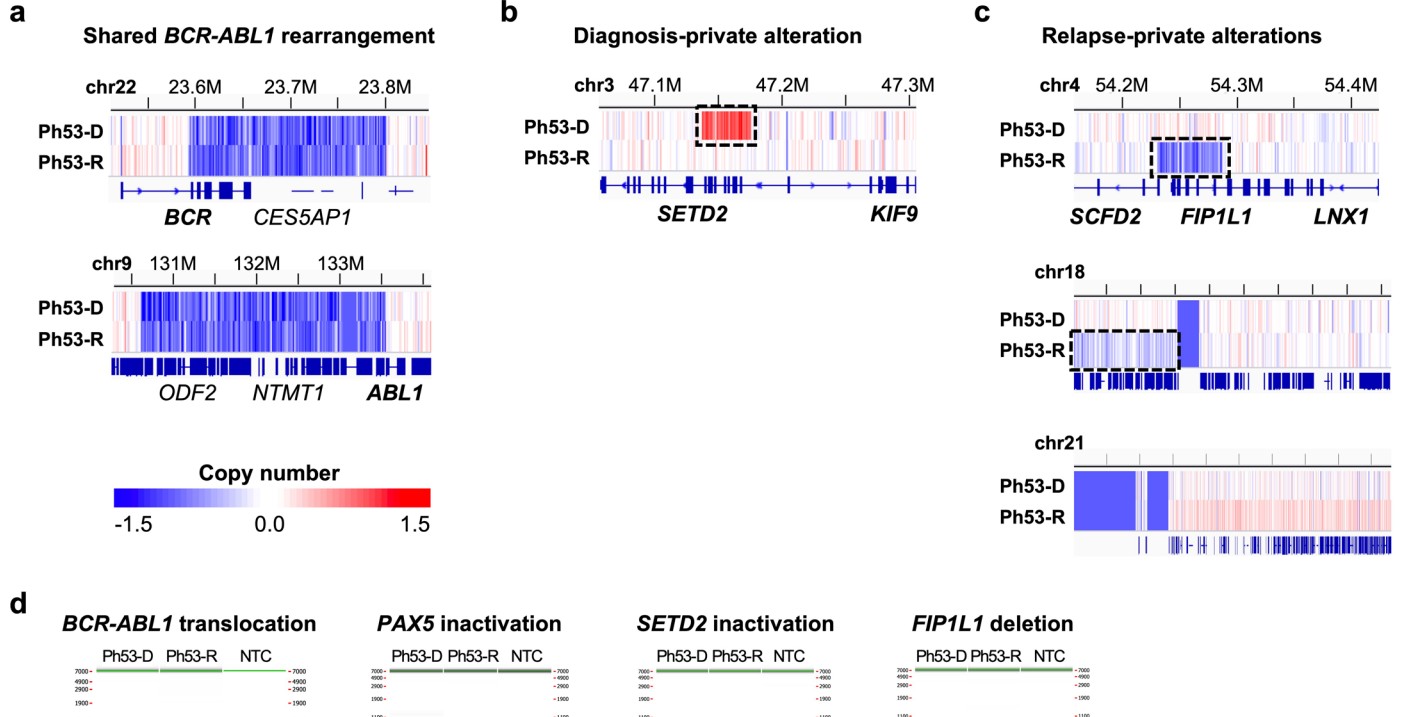

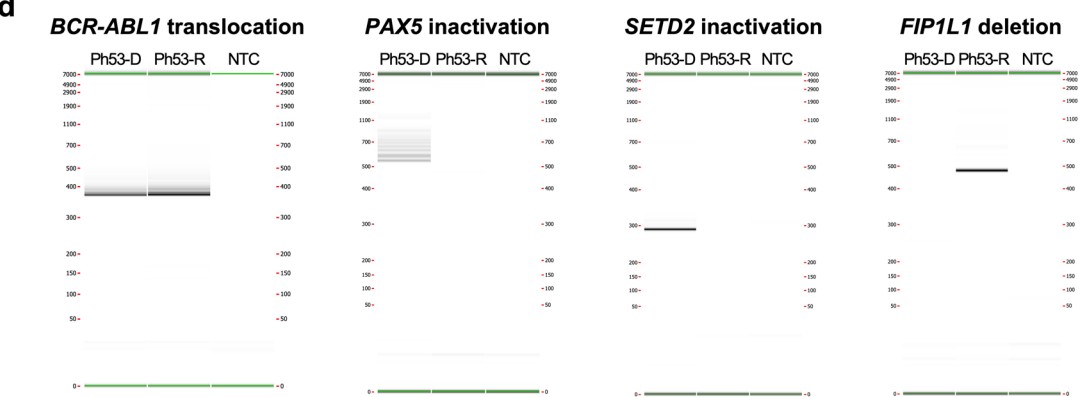

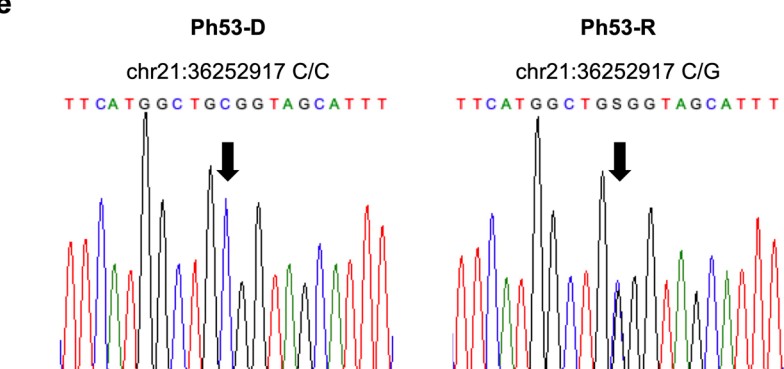

**Extended Data Fig. 8 | Private and shared alterations in Ph53 at diagnosis and relapse. a**, Shared *BCR-ABL1* translocation between Ph53-D and Ph53-R as evidenced by identical losses of flanking regions. **b**, *SETD2* inactivation is private to Ph53-D. **c**, *FIP1L1* deletion, gain of 18p, and trisomy 21 are private to Ph53-R. In A-C, blue represents copy number loss and red represents copy number gain relative to the reference T-cell genome. **d**, Nested PCR validation of SVs in Ph53.

*BCR-ABL1* translocation is shared, *PAX5* and *SETD2* inactivations are private to Ph53-D, and *FIP1L1* deletion is private to Ph53-R. NTC, no template control. **e**, Sanger sequencing validation of C>G substitution at chr21:36252917 (*RUNX1* c.445G>C, p.A149P) in Ph53-R (right). This mutation is not detected in Ph53-D (left).

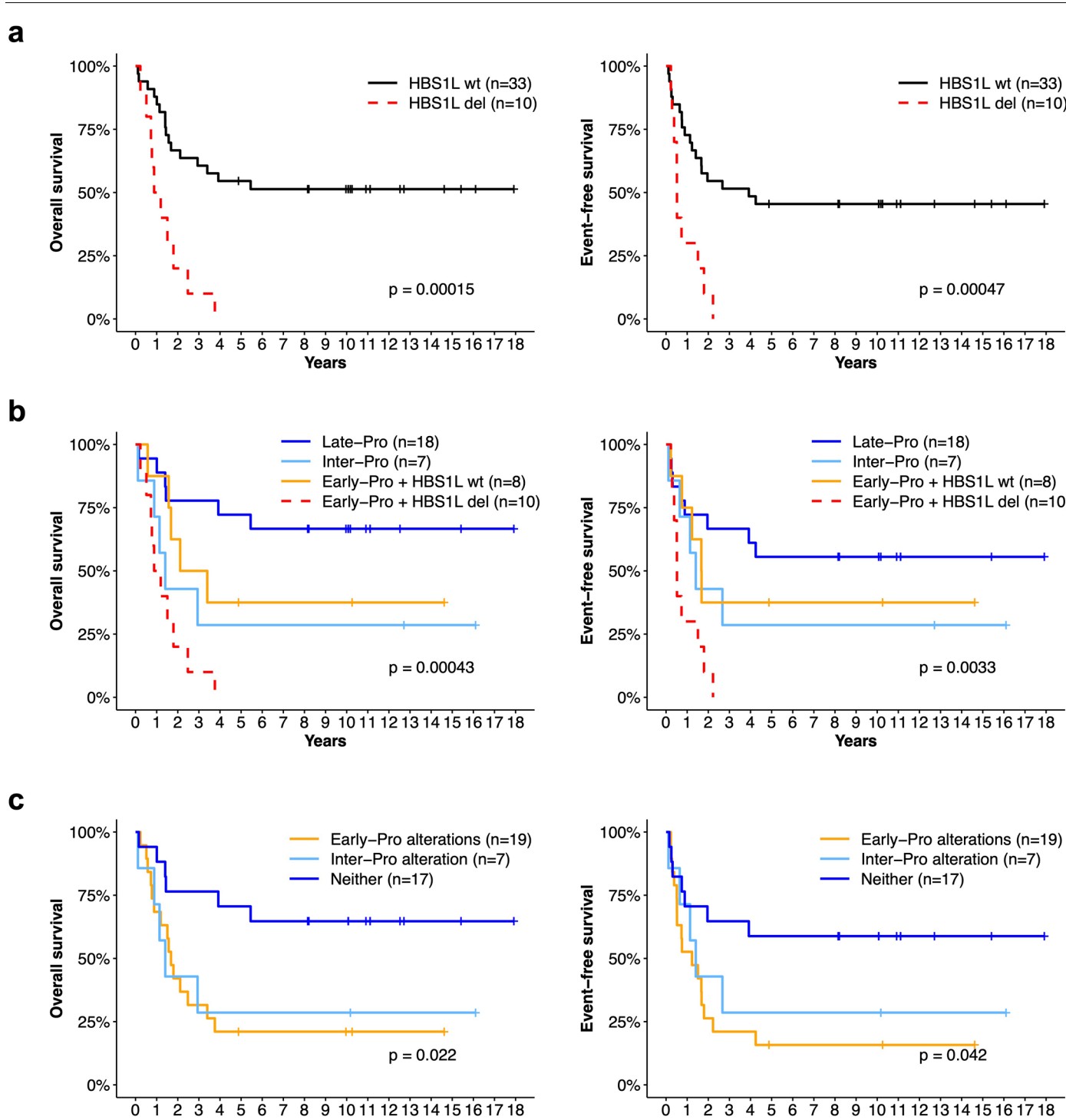

**Extended Data Fig. 9 | Survival analysis by *HBS1L* status and by subtype-specific genomic alterations. a**, Survival analysis of patients grouped by *HBS1L* status (wildtype vs. deleted). **b**, Survival analysis of patients grouped by *HBS1L* status and molecular subtypes. Late-Pro and Inter-Pro patients are all *HBS1L* wildtype. **c**, Survival analysis of patients grouped by subtype-specific genomic

alterations. Early-Pro alterations are *HBS1L*, *RUNX1*, *EBF1*, and/or Monosomy 7, and Inter-Pro alteration is two-copy loss of *IKZF1*. Kaplan-Meier estimates of overall survival (left) and event-free survival (right) are shown. Only the main cohort (n = 43 for survival analysis) is analyzed since these patients have corresponding genomic data. wt, wildtype; del, deleted.

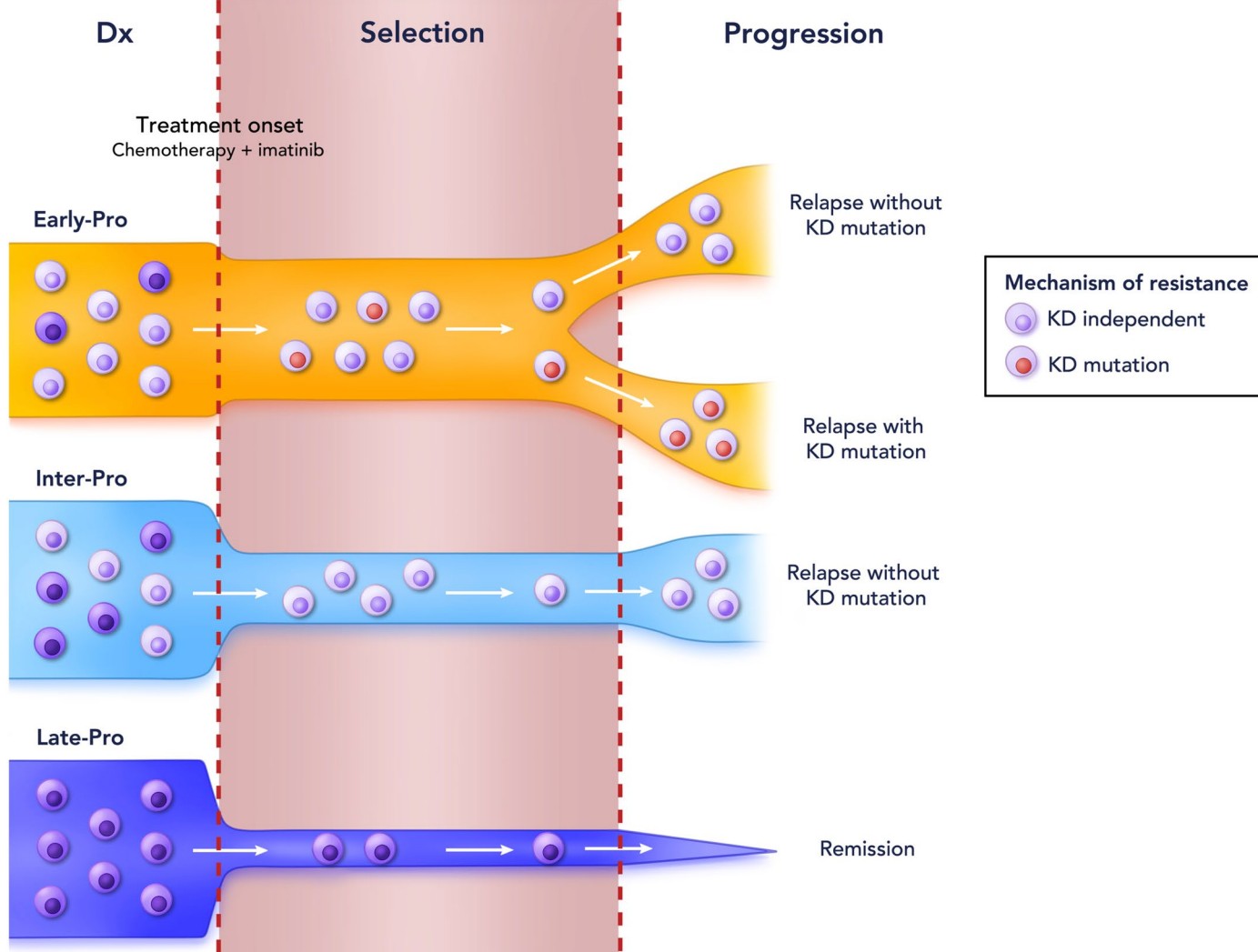

**Extended Data Fig. 10 | Treatment resistance mechanisms in *BCR-ABL1* lymphoblastic leukemia.** The degree of innate resistance to chemotherapy and TKI (that is imatinib) influences the probability of acquiring *BCR-ABL1* KD mutations and relapsing. Late-Pro leukemias respond extremely well to induction and most patients achieve remission/MMR. Inter-Pro leukemias do not respond as well and most patients relapse without KD mutations. Early-Pro leukemias show minimal response to induction and most patients relapse with or without KD mutations. Cell quiescence, STAT5 signaling, and UPR signaling potentially contribute to treatment resistance in the Early-Pro and Inter-Pro subtypes.

# Reporting Summary

## Statistics

For all statistical analyses, confirm that the following items are present in the figure legend, table legend, main text, or Methods section.

| n/a | Confirmed | |
|---|---|---|
| ☐ | ☒ | The exact sample size (*n*) for each experimental group/condition, given as a discrete number and unit of measurement |
| ☐ | ☒ | A statement on whether measurements were taken from distinct samples or whether the same sample was measured repeatedly |
| ☐ | ☒ | The statistical test(s) used AND whether they are one- or two-sided<br>*Only common tests should be described solely by name; describe more complex techniques in the Methods section.* |
| ☐ | ☒ | A description of all covariates tested |
| ☐ | ☒ | A description of any assumptions or corrections, such as tests of normality and adjustment for multiple comparisons |
| ☐ | ☒ | A full description of the statistical parameters including central tendency (e.g. means) or other basic estimates (e.g. regression coefficient) AND variation (e.g. standard deviation) or associated estimates of uncertainty (e.g. confidence intervals) |
| ☐ | ☒ | For null hypothesis testing, the test statistic (e.g. *F*, *t*, *r*) with confidence intervals, effect sizes, degrees of freedom and *P* value noted<br>*Give P values as exact values whenever suitable.* |
| ☒ | ☐ | For Bayesian analysis, information on the choice of priors and Markov chain Monte Carlo settings |
| ☐ | ☒ | For hierarchical and complex designs, identification of the appropriate level for tests and full reporting of outcomes |
| ☐ | ☒ | Estimates of effect sizes (e.g. Cohen's *d*, Pearson's *r*), indicating how they were calculated |

*Our web collection on statistics for biologists contains articles on many of the points above.*

## Software and code

Policy information about availability of computer code

| Data collection | Not applicable |
|---|---|
| Data analysis | R v3.6.1, STAR v2.5.2a, HTSeq v0.9.1, DEseq2 v1.18.1, sva v3.42.0, ConsensusClusterPlus v1.40.0, GSEA Prerank v4.1.0, STAR-Fusion v1.0.0, MiXCR v3.0.3, CellRanger v3.0.1, Seurat v3.1.2, SingleR v1.0.6, Monocle 3 v0.2.3.0, Symphony v0.1.1, bwa v0.7.12, Picard v1.90, Genome Analysis Tool Kit (GATK) v3.6.0, Strelka v1.0.7, MuTect v1.1.4, HMMcopy v0.1.1, CREST alpha, Delly v0.5.5, Debarcer, COMPASS v4.0.0, MaxQuant v1.6.3.3, and WebLogo v2.8.2 |

For manuscripts utilizing custom algorithms or software that are central to the research but not yet described in published literature, software must be made available to editors and reviewers. We strongly encourage code deposition in a community repository (e.g. GitHub). See the Nature Portfolio guidelines for submitting code & software for further information.

## Data

Policy information about availability of data

All manuscripts must include a data availability statement. This statement should provide the following information, where applicable:
- Accession codes, unique identifiers, or web links for publicly available datasets
- A description of any restrictions on data availability
- For clinical datasets or third party data, please ensure that the statement adheres to our policy

Whole genome sequencing and RNA-seq data are deposited and publicly available on the European Genome-Phenome Archive (EGAS00001007167). Mass spectrometry data can be found in Supplementary Table 14.

## Human research participants

Policy information about studies involving human research participants and Sex and Gender in Research.

| | |
|---|---|
| Reporting on sex and gender | Biological sex is used throughout the paper and sex proportions of cohorts and leukemia subtypes are provided. In 93 patients, 53 were male and 40 were female. |
| Population characteristics | In the combined cohort, 93 diagnostic (treatment-naive) samples and 4 relapse samples were studied. Median age of the cohort is 51 y.o. (range 13~88 y.o.). 86 patients were diagnosed with B-cell acute lymphoblastic leukemia, 5 with chronic myeloid leukemia in lymphoid blast crisis, and 2 with mixed-phenotype acute leukemia. |
| Recruitment | We studied BCR-ABL1 lymphoblastic leukemia samples from the UHN Leukemia Biobank in Toronto, Canada. The first 27 samples were selected to have equal proportions of p190 and p210 isoforms of BCR-ABL1 fusion, and the remaining samples were randomly selected. Because the initial observation, and part of the final conclusion, of this study was that transcriptomic classes of BCR-ABL1 ALL are not driven by BCR-ABL1 isoforms, we believe this initial selection process has no impact on the results. This is also mentioned in Methods. |
| Ethics oversight | The study was approved by the Research Ethics Board of the University Health Network (REB# 01-0573) and written informed consent was obtained from all patients. |

Note that full information on the approval of the study protocol must also be provided in the manuscript.

# Field-specific reporting

Please select the one below that is the best fit for your research. If you are not sure, read the appropriate sections before making your selection.

☒ Life sciences  ☐ Behavioural & social sciences  ☐ Ecological, evolutionary & environmental sciences

For a reference copy of the document with all sections, see nature.com/documents/nr-reporting-summary-flat.pdf

# Life sciences study design

All studies must disclose on these points even when the disclosure is negative.

| | |
|---|---|
| Sample size | Samples from 93 patients with BCR-ABL1 lymphoblastic leukemia were obtained from the University Health Network Leukemia Biobank, who collected them between 1992 and 2019. This disease is rare, and based on our initial observation of two broad phenotypes - mixed-lineage and B-cell precursor - we deemed the sample size sufficient. Sample sizes for single-cell RNA-seq (n=9), colony formation assay (n=14), and mass spectrometry (n=16) were determined by availability of resources and frozen cells/pellets. For these experiments, every effort was made to maintain the observed proportions of leukemia subtypes. |
| Data exclusions | All patient samples were included in analyses except for the following. Survival analyses were performed after excluding 12 out of 93 patients; 5 were CML cases in lymphoid blast crisis, 5 were not given a TKI at induction, and 2 died within 30 days of diagnosis. For analysis of clinical flow cytometry data, most samples had antigen expression data available, but a small subset did not have data available for certain antigens. |
| Replication | To confirm our findings, we investigated two patient cohorts through different experimental approaches. Samples from the main cohort (n=53) were purified by FACS and subjected to whole-transcriptome sequencing, whereas samples from the second cohort (n=40) were purified by magnetic separation and subjected to 3' RNA-seq. Transcriptomic clusters identified from the main cohort were replicated in the second cohort. Subsequent experiments, such as single-cell RNA-seq and colony formation assay, were performed with multiple samples from each leukemia subtype when possible (i.e. biological replicates) but were not repeated. |
| Randomization | In this retrospective study, we randomly selected BCR-ABL1 lymphoblastic leukemia samples stored at the University Health Network Leukemia Biobank without any prior knowledge regarding sex, age, treatment responses, outcomes, flow cytometry findings, or molecular findings (except for detection BCR-ABL1) of the patients. Data from this study were not used to alter the clinical treatment of the patients. |

| Blinding | This is not relevant to our study as there was no group allocation. After molecular subtypes were identified, sample IDs were known to the investigators to keep track of samples used in each experiment/analysis. |
|---|---|

# Reporting for specific materials, systems and methods

We require information from authors about some types of materials, experimental systems and methods used in many studies. Here, indicate whether each material, system or method listed is relevant to your study. If you are not sure if a list item applies to your research, read the appropriate section before selecting a response.

## Materials & experimental systems

| n/a | Involved in the study |
|---|---|
| ☐ | ☒ Antibodies |
| ☒ | ☐ Eukaryotic cell lines |
| ☒ | ☐ Palaeontology and archaeology |
| ☒ | ☐ Animals and other organisms |
| ☒ | ☐ Clinical data |
| ☒ | ☐ Dual use research of concern |

## Methods

| n/a | Involved in the study |
|---|---|
| ☒ | ☐ ChIP-seq |
| ☐ | ☒ Flow cytometry |
| ☒ | ☐ MRI-based neuroimaging |

## Antibodies

| Antibodies used | Anti-phosphorylated-STAT5 (C11C5; Cell Signaling Technology #9359; 1:100 dilution), PE anti-CD19 (4G7; BD Biosciences #349209; 4uL/10^6 cells), APC-eFluor 780 anti-CD34 (4H11; eBioscience #47-0349-42; 4uL/10^6 cells), eFluor 450 anti-CD45 (2D1; eBioscience #48-9459-42; 5uL/10^6 cells), Super Bright 645 anti-CD3 (OKT3; eBioscience #64-0037-42; 3.5uL/10^6 cells), PerCP-eFluor710 anti-CD34 (4H11; eBioscience #46-0349-42; 4uL/10^6 cells), PE-Cy7 anti-CD38 (HIT2; BD Biosciences #560677; 3.5uL/10^6 cells), APC anti-CD90 (5E10; BD Biosciences #559869; 5uL/10^6 cells), FITC anti-CD45RA (HI100; BD Biosciences #555488; 5uL/10^6 cells), PE-Cy5 anti-CD49f (GoH3; BD Biosciences #551129; 4.5uL/10^6 cells), AlexaFluor700 anti-CD10 (CB-CALLA; eBioscience #56-0106-42; 5uL/10^6 cells), biotin anti-FLT3 (4G8; custom conjugation from BD Biosciences; 8uL/10^6 cells) and Qdot605 streptavidin (Thermo Fisher Scientific #Q10101MP; 3uL/10^6 cells) |
|---|---|
| Validation | We used commercially available antibodies from Cell Signaling Technology, BD Biosciences, and Thermo Fisher Scientific that were validated to bind human antigen targets. |

## Flow Cytometry

### Plots

Confirm that:

☒ The axis labels state the marker and fluorochrome used (e.g. CD4-FITC).

☒ The axis scales are clearly visible. Include numbers along axes only for bottom left plot of group (a 'group' is an analysis of identical markers).

☒ All plots are contour plots with outliers or pseudocolor plots.

☒ A numerical value for number of cells or percentage (with statistics) is provided.

### Methodology

| Sample preparation | Samples consisted of primary human leukemia biopsies from peripheral blood (n=85) and bone marrow (n=8). Samples were thawed, labelled with antibodies, and flow-sorted or separated via magnetic beads. |
|---|---|
| Instrument | BD FACSAria III |
| Software | BD FACS Diva was used for cell sorting and BD FlowJo was used for post-sort analysis. |
| Cell population abundance | 5~10 million cells from bulk leukemia biopsy samples were used for flow sorting. Leukemic blasts accounted for >70% in most cases. |
| Gating strategy | 1) FSC-A vs. SSC-A (both medium to high); 2) FSC-H vs. FSC-W (FSC-W low); 3) SSC-H vs. SSC-W (SSC-W low); 4) FSC-A vs. PI (PI low); 5) CD34, CD19, and sometimes CD45 were used to identify and sort blasts. |

☒ Tick this box to confirm that a figure exemplifying the gating strategy is provided in the Supplementary Information.

