## [Peer Review File · Nature Genetics]

Peer Review Information

Manuscript Title: Transcriptomic classes of BCR-ABL1 lymphoblastic leukemia

Corresponding author name(s): Dr Faiyaz Notta

Reviewer Comments & Decisions:

Decision Letter, initial version:

23rd Sep 2022

Dear Dr Notta,

First of all, please accept my sincere apologies for the delay in returning this decision to you. Thank you so much for your patience.

Your Article, "Transcriptomic classes of BCR-ABL1 lymphoblastic leukemia" has now been seen by 2 referees. As Reviewer #3 did not respond to changes, we chose to move forward without their feedback.

You will see from their comments copied below that while they find your work of considerable potential interest, they (particularly Reviewer #2) have raised quite substantial concerns that must be addressed. In light of these comments, we cannot accept the manuscript for publication, but would be very interested in considering a revised version that addresses these serious concerns.

We hope you will find the referees' comments useful as you decide how to proceed. If you wish to submit a substantially revised manuscript, please bear in mind that we will be reluctant to approach the referees again in the absence of major revisions. We ask that you address all the reviewer points in full - experimentally where possible - or textually where appropriate.

If you choose to revise your manuscript taking into account all reviewer and editor comments, please highlight all changes in the manuscript text file. At this stage we will need you to upload a copy of the manuscript in MS Word .docx or similar editable format.

*2) If you have not done so already please begin to revise your manuscript so that it conforms to our Article format instructions, available [here](http://www.nature.com/ng/authors/article_types/index.html). Refer also to any guidelines provided in this letter.

[redacted]

If you wish to submit a suitably revised manuscript we would hope to receive it within 6 months. If you cannot send it within this time, please let us know. We will be happy to consider your revision so long as nothing similar has been accepted for publication at Nature Genetics or published elsewhere. Should your manuscript be substantially delayed without notifying us in advance and your article is eventually published, the received date would be that of the revised, not the original, version.

Thank you for the opportunity to review your work.

Sincerely,

Safia Danovi
Editor
Nature Genetics

Referee expertise:

Referee #1: transcriptomics

Referee #2: leukemia

Reviewers' Comments:

Reviewer #1:

Remarks to the Author:

The manuscript entitled "Transcriptomic classes of BCR-ABL1 lymphoblastic leukemia", describes how BCR-ABL1 lymphoblastic leukemia can be subtyped into three groups based on gene expression. Kim et al. associate each with distinct stages of B-cell differentiation, sets of mutations and treatment response. They hypothesize that BCR-ABL1 lymphoblastic leukemia originates through alterations at two stages: first a BCR-ABL1 fusion arising in stem cells followed by transformation induced by a second set of mutations in B-cell progenitors. This is supported by evidence of SVs arising from RAG recombination and TdT being active for the second set of alteration, but not BCR-ABL1. Suggesting they occur in lymphoid differentiation.

I found this to be a very interesting manuscript and an important contribution to characterizing BCR-ABL1 lymphoblastic leukemia. The authors provide good data to support their conclusions and the manuscript is very well written, with appropriate figures. I only have a few suggestions to improvement the manuscript:

Major comments:

1. The model put forward by the authors (Fig. 4a_{ii}) speculates that BCR-ABL1 arises in stem cells prior to lymphoid differentiation, and there is evidence of this from BCR-ABL1+ myeloid colonies in 29% of their patients. However, it was not clear to me whether the authors were also trying to exclude the alternative model (Fig. 4a_i), where BCR-ABL1 occurs later e.g. "The findings align with the notion that BCR-ABL1 translocation originates in multipotent cell types". The data points to both being possibilities, e.g. patients where myeloid colonies were not identified, 10% of BCR-ABL1 SVs with an RSS motif. Although I found the RSS and TdT analysis striking and supportive of the timing of transforming mutations, using this to infer the timing of BCR-ABL1 seems weaker. Association between RSS motifs and TdT related insertions are very likely to be SV specific (Fig 3e). Moreover, can translocation through other mechanisms at the lymphoid stage be excluded? Some clarification in the manuscript text should be provided.
2. If feasible the transcriptomic groups (and associated mutations) should be confirmed using other available cohort data such as from TARGET and/or St. Jude. As these include pediatric samples, it would be informative to see if the findings also apply to pediatric BCR-ABL1 ALL.
3. The Inter-Pro subtype has a mutational profile extremely similar to Late-Pro (Fig. 3a), but with a

double hit in IKZF1. Some more discussion and exploration of this would be valuable. Does a second hit in IKZF1 of the Late-Pro group transform it and its transcriptomic profile into Inter-Pro? Is there any evidence of subclonal single hit IKZF1 (e.g. in the single cell or whole genome) data from the Inter-Pro sample? Or double hit IKZF1 in subclones of the Late-Pro patients? This may be technically hard to identify from the data, but could be interesting if found.

Minor comments:

- The bars showing proportions of each cell type in Fig 2f are very effective at visualizing differences between cell types across samples and I felt that Fig 2g would profit from being replaced with this style of visualization. Importantly, this would allow the variation between patients of the same subtype (replicates) to be seen. Interquartile ranges is not really sufficient for this.
- BCR-ABL1 is known to be detected at low levels in healthy individuals (e.g. in GTEx <https://www.biorxiv.org/content/10.1101/2021.08.02.454639v2.full>). As this supports the authors hypothesis that cell of origin and cell of transformation are different it could be worth adding to the discussion.
- Are there any implications from these findings for BCR-ABL1+ CML?

Reviewer #2:

Remarks to the Author:

Transcriptomic classes of BCR-ABL1 lymphoblastic leukemia

In this manuscript Kim JC... Notta F et al have used molecular profiling of BCR-ABL+ adult ALL to identify transcriptional heterogeneity that might explain treatment resistance and relapse, in the context of KD mutations and without. The manuscript is well written and the presentation is very clear.

I read the manuscript with interest and commend the authors for the work done to try and address a very relevant clinical question in BCR-ABL+ ALL as there is a clear need for identifying the molecular mechanisms which lead to treatment failure. I particularly commend their use of primary patient material for these assays, including matched transcriptomic and WGS data, and correlating findings with clinical outcome data.

Previous literature has shown that BCR-ABL translocation in ALL might arise in progenitors upstream of CD34+CD19+ cells, and molecular heterogeneity at a stem cell level has been shown for BCR-ABL+ CML; these need to be acknowledged and discussed.

Summary of key findings:

1) There were 3 distinct transcriptomic subtypes of BCR-ABL+ ALL in a cohort of 57 patients, which was validated in a 2nd cohort of 40 patients (named C1, C2 and C3). C1 had expression of non-lymphoid and stem cell genes, was more quiescent, whereas C3 had the most mature B cell gene expression and were the least quiescent. This also correlated with cell surface expression of stem, myeloid and B-cell markers (by flow cytometry) in these clusters. 2) The GEP of ALLs in each cluster was then compared to that of normal flow sorted cord blood HSPC: C1 patients clustered with normal 'early ProB', C2 with 'pre-proB' and C3 with 'ProB'. Here I have concerns about the nomenclature and the normal B cell hierarchy described, in particular Pre-proB and ProB cells. 3) The authors perform single cell gene expression profiling by 10x Chromium for just under 30,000 leukemic cells from 9 patients which corroborates findings in (2). 4) Specific secondary genomic transformation events were

enriched in the 3 different subtypes of BCR-ABL+ ALL with EBF1 mutations being predominant in C1 and PAX5, CDKN2A/B and RB1 deletions in C3. The genes affected appear to correlate with GEP of the B-cell differentiation stage the leukemic cells resemble. 5) Secondary genetic events were more strongly associated with RSS motifs and active TdT than the BCR-ABL translocation, suggesting they occurred in a more B-lymphoid committed cell than the translocation itself. While this is an interesting hypothesis, it is still speculative as it may just mean that the translocation is not RAG machinery dependent. 6) Colony forming assays from upstream progenitors: this is probably stronger proof that the translocation occurs upstream of CD19+ and other lymphoid progenitors than the data presented in (5). Patient CD34+CD19-CD45RA- cells (these would include HSC, MPP, CMP and MEP, but exclude LMPP, MLP and GMP) gave rise to myeloid colonies that were tested for BCR-ABL and secondary events. 4/14 patients' cells were positive for BCR-ABL in the colonies generated; but none had secondary genetic events. 7) The last section correlates clinical outcomes with the 3 molecular subtypes showing early ProB type has poorer prognosis in terms of survival and TKI response/resistance, and early ProB and inter-ProB ALLs benefit from 2nd generation TKI.

Methodology: They have used bulk RNA-sequencing (2 cohorts, total 93 patients), single cell RNA sequencing (9 patients), whole genome sequencing (57 patients) and functional clonogenic assays (14 patients) to assess the transcriptomic and genomic subtypes of BCR-ABL+ ALL in adults. They also correlated these subtypes with patient outcome and treatment resistance where clinical data was available. The methodology is robust, where there are concerns, including in data interpretation/sample size or statistics; I have raised them under relevant sections in comments below.

Specific comments:

Results:

1) Distinct transcriptomic clusters of BCR-ABL1 lymphoblastic leukemia

Comments:

- the selection method for blasts (CD19 bead selection) was different for cohort 2; and depending on blast %, might be less pure than the original cohort and could skew the transcriptomic profile; this needs to be acknowledged in methods or text and/or CD19+ blast % could be added to Table S2. Especially as some of the samples where blasts were sorted (6 different types according to Fig S35A) have either excluded CD34-CD19+ cells or have a large proportion of CD34+CD19-/lo cells that would not be represented in cohort 2.
- The authors claim that the 3 clusters were not associated with the type of leukemia, but the no. of patients in MPAL and CML-BC groups is probably too small to comment about their distribution in the 3 clusters
- C1 cluster showed the highest expression of CD34 gene: in some instances, the cells were specifically sorted as CD34+CD19+ so this is to be expected if these samples make up majority of C1. In some cases (blast Type 2), there is a large proportion of cells that are CD34+CD19-, how can the authors be sure these are blasts? Are the C1 samples skewed by such sample sorting? Is the CD34 and stem cell/myeloid signature also the case in the 2nd cohort where cells were CD19 selected only? It would be good to see Fig S9 and S10 separately for the 2 cohorts of patients as the composition of cells selected are slightly different
- The authors comment that C1 patients express CD13/33 more frequently and have more blasts expressing cyMPO despite being below the diagnostic threshold of 10% blasts being cyMPO+ to diagnose MPAL, therefore RNAseq might be better at picking up MPALs which might be missed by flow. Did all the C1 patients with <10% cyMPO also express CD13 and CD33? CD13 and CD33 are often

aberrantly expressed in many B-ALLs which don't have any myeloid lineage potential. Although we don't know whether flow techniques are missing cyMPO in these cells, low level gene expression may not necessarily mean protein expression. So, while gene expression data is useful information, I would be wary of the diagnostic benefit of bulk RNAseq over flow cytometry, which is essentially checking for protein expression at single cell level. It is a very useful measure in experienced hands. This is also evident in C2 samples- where MPO gene expression levels are similar to C1, but cyMPO is not seen. Could the authors show a correlation graph for MPO expression and cyMPO in the patients where both are available?

2) Each transcriptomic cluster aligns to a specific stage of normal B-cell development

- Please provide the flow sorting data for the normal CB populations being used as the normal reference template (Table S5). How many samples were analysed (n=4 from Fig 2), details of cell numbers per population and methodology for RNAsequencing? Was it adapted for small cell numbers?
- The nomenclature used for the normal B cell hierarchy in this paper has been adapted from Hystad et al, 2007 ABM microarray data. There are many publicly available RNA-seq datasets of normal ABM HSPC, did the authors try using these? E.g: (<https://www.nature.com/articles/ng.3646>)
- Mapping the patient samples onto available normal ABM datasets would be more appropriate than cord blood (including single cell data from Human Cell Atlas). Could the authors provide this data? E.g: <https://www.ncbi.nlm.nih.gov/pmc/articles/PMC6296228/>
- The ProB population in the Hystad paper (used as a reference) is CD34+CD19+CD10+IgM-. The ProB population described in this paper is rather confusing as it is described as CD34+CD10-CD19+ while being downstream of a CD34+CD10+CD19+ PreProB cell and upstream of a CD34-CD19+CD10+ PreB cell. This does not normally happen in B cell development: the acquisition and loss of CD10 in the CD34 compartment and then gain of the marker again after CD34 loss. Usually CD10 comes on in a CD34+CD38+ progenitor (CLP- labelled early ProB in this paper: CD34+CD10+CD19-) and then stays on till it is lost in a naïve B cell (CD34-CD19+CD10-IgM+IgD+). Also note that the CD34+CD10-CD19+ population has only been described as part of an alternate pathway in fetal life and CB; and doesn't lie downstream of the 34+19+10+ progenitor, but upstream of the ProB progenitor (CD34+CD10+CD19+)- proven both functionally and molecularly (<https://pubmed.ncbi.nlm.nih.gov/12446447/>; <https://pubmed.ncbi.nlm.nih.gov/20231472/>; <https://pubmed.ncbi.nlm.nih.gov/31383639/>).

Although the marker genes for these normal CB populations described are not available to us (this data should be made available); the C3 cluster which corresponds to the ProB population has both expression of CD10 and MME which is contradictory to the definition of the normal counterpart. It also has a BCR signalling and expresses IGK and IGL which makes it either a PreB cell or lower in the B cell hierarchy (according to the author's own hierarchy scheme in Fig 2e).

- It is also unusual to have CD10-CD20+IgM- PreB cells. CD10 is usually lost after both IgM and IgD are expressed on a cell (transitional B cells). Could the authors reconcile these points?
- While I do not question that C1, C2 and C3 have gene expression profiles in order of increasing B cell differentiation, (this is very evident from the gene expression and flow data), the normal reference dataset needs clarifying especially the preproB and ProB cell types. Could you show the hierarchical clustering of the normal cells separately and that the B cell gene expression programs change in the expected pattern?
- I would be cautious about concluding that the 3 clusters represent a block in differentiation at different stages of B cell development. The results simply show that they are molecularly similar to different lymphoid/B cell progenitors. As almost all the ALLs (in all 3 clusters) are CD19+ and have IgH rearrangement, this implies that they must have differentiated till at least a ProB/PreB stage.

3) Early-Pro leukemias display significant hematopoietic lineage plasticity

It is not very clear what cells were used and how these were selected (were they all CD19+?), before loading on the chip. The sc dataset has not been exploited enough in my opinion; other than again correlating to normal CB cell types and demonstrating that C1 expresses lympho-myeloid genes (it doesn't really add to what was already shown in (2)).

- In samples where there was a high frequency of CD34+19lo/- cells e.g Ph4-D how can the authors be sure that the cells mapping to GMP, LMPP etc are blasts (i.e BCR-ABL+) and not normal myeloid progenitors? Were any patients from the cohort 2 where CD19 selected? It would be interesting if they had a stem like phenotype that mapped to immature progenitors.
- Could the authors show gene expression data on the UMAPs for some key stem cell, myeloid and B cell genes (especially CD19)? As well as cell cycle scores.
- The single cell data could also be interrogated to ascertain what % of cells are BCR-ABL fusion transcript+ve and whether they mapped only to CD19+ cells.
- Diffusion maps/FDG would be useful for the patient samples either individually or all 9 patient datasets could be combined to determine whether early ProB ALLs (C1) indeed lie upstream of others.

4) Distinct transformation events define each molecular subtype

- While this data is very interesting, the exclusivity of some mutations being present in particular clusters should be interpreted with caution because of the small numbers of patients with these second hits.
- 3 of the ALLs with Trisomy 21 (n=5) are also hyperdiploid. Is the T21 part of the hyperdiploidy?
- How do these results help patient risk stratification or treatment?

5) DNA footprints of lymphoid enzyme activity inform on timing of transformation events

- Were RAG1, RAG2 and DNTT expressed in C1, C2 and C3? And at a sc level in the 10x dataset? This data would be nice to include to support the hypothesis that secondary genetic events are RAG mediated.
- The authors propose an interesting hypothesis that because the DNA footprints of lymphoid enzyme activity is associated more with genetic events other than the BCR-ABL translocation itself, these secondary hits and ALL transformation occur in downstream B lymphoid cells as opposed to the primary translocation that occurs in a more primitive progenitor, but this is speculative. The BCR-ABL translocation may just not be RAG mediated rather than occurring in a cell before RAG and TdT is expressed. I would tone down this conclusion based on these results alone.

6) Transformation events do not accumulate in the cell-of-origin where BCR-ABL1 occurred

- It is unclear whether these were single cell colony forming assays (as stated in figure titles and methods); i.e were single CD34+19-45RA- cells sorted into wells and checked for colony formation and resulting colonies genotyped? If so, were they index sorted so the identity of the cell giving rise to a BCR-ABL+ colony can be determined? For e.g an HSC or an MPP? If not, how many cells were plated/ sample in the 3.5cm petri dish? This is bulk colony forming assay and the figure titles and methods heading should be amended, as they are confusing.
- What was the clonogenicity of each patient sample? Was there much variation in total number of colonies generated?
- It is interesting that no secondary genetic events were detected in any of the BCR-ABL+ colonies. The authors suggest this indicates that the cell-of-origin in which BCR-ABL1 occurred did not accumulate genetic events related to transformation. This could be an over interpretation given 21-87

colonies/patient were being tested of which 3, 3, 11 and 34 colonies were BCR-ABL+ in 4 patients. A minor BCR-ABL+ clone with additional mutation might easily have been missed. Or that cell type might have not survived in these culture conditions. This can only be definitively answered by sc genotyping of HSC/MPP and downstream progenitors from patients, that looks for clonal hierarchy. This should be discussed, along with the results from previous studies (<https://pubmed.ncbi.nlm.nih.gov/15735032/>, <https://pubmed.ncbi.nlm.nih.gov/15908956/>).

7) Cellular stage of leukemic transformation impacts treatment outcome in patients
 - Fewer earlyProB ALLs achieved MMR after induction and DMR was only achieved in Late ProB ALLs and this translated to variable OS and EFS in the 3 subtypes
 - Could the authors speculate why Late ProB ALLs don't benefit from newer TKIs? Fig 5d: these are a handful of patients in each arm, and I don't think any conclusions can really be drawn from this data. Suggest remove.

8) Innate resistance to chemotherapy and TKIs in Early-Pro and Inter-Pro leukemias
 The primary question that the authors set out to address was how the molecular profile can be used in the clinical context to explain BCR-ABL+ ALL relapse that occurs without KD mutations.
 - The authors suggest that cell quiescence and KD-independent resistance mechanisms such as STAT5 signalling are more prevalent in early ProB and inter-ProB ALLs which could be counteracted by more potent TKIs.

Conclusions: Could the authors elaborate how clinical decision making and treatment plans could be altered based on all of their findings in discussion section in bit more detail.
 Is it possible that these molecular clusters are driven by levels of CD34/CD19 expression on blasts, and much of this could be resolved just by immunophenotype, i.e CD34+CD19lo BCR-ABL+ ALLs need a rethink. This would make it much easier for quick turnaround and decision making in the clinical setting.

MINOR COMMENTS:

- Methods: FACS sorting: 'variable numbers of HSC, MPP, MLP, CMP, GMP, MEP, and mature B-cells were also collected when possible'. Where is this data shown in the paper? Gating strategy? What cell numbers?
- Methods: Non-negative matrix factorization and consensus hierarchical clustering: What were the 163 genes used for consensus clustering. What were the marker genes for the 3 clusters? Please provide a full list as supplementary tables
- Methods: Single-cell RNA-seq and phenotype annotation: how were the 10,000 cells/sample selected? Reference dataset: what cells are these? CB? ABM? FACS sorted for bulk RNAseq? This data needs to be provided unless available elsewhere as a publication
- Methods: Single cell colony assays: please clarify what this means: were these were sorted as single cells. If not, how many cells were sorted and plated?
- Figure 4b: check label of Ph14-D – should read 'early ProB'?
- The method for deriving the phylogenetic relationship using variants between diagnosis and relapse samples (Fig 4d) needs a little bit more detail
- Please ensure all figures appear in the right order and are mentioned in the text (including suppl figures).

Author Rebuttal to Initial comments

[insert PDF]

Decision Letter, first revision:

18th Jan 2023

Dear Dr. Notta,

Thank you for submitting your revised manuscript "Transcriptomic classes of BCR-ABL1 lymphoblastic leukemia" (NG-A60593R). It has now been seen by the original referees and their comments are below. The reviewers find that the paper has improved in revision, and therefore we'll be happy in principle to publish it in Nature Genetics, pending minor revisions to satisfy the referees' final requests and to comply with our editorial and formatting guidelines.

Sincerely,

Safia Danovi
Editor
Nature Genetics

Reviewer #1 (Remarks to the Author):

I thank the authors for their very comprehensive response to each of my comments. Their new results support the original conclusions and the observation of the three subtypes in Ph-like B-ALL is quite interesting. They have addressed each point to my satisfaction and I have only one further (minor) suggestion for improvement.

In the abstract, the language used suggests that the relationship between subtypes and B-cell maturation stage is proven, e.g. "A later arrest..". Could this be rephrased to be less definitive. e.g. "The subtype associated/consistent with a later arrest..."

Reviewer #2 (Remarks to the Author):

Transcriptomic classes of BCR-ABL1 lymphoblastic leukemia

Thank you for providing a revised version of this manuscript. My comments and concerns have been addressed adequately, and I commend the authors for providing valuable additional data where required. The manuscript is much improved, many congratulations.

I only have a few minor comments detailed below (not all require a response from the authors!)

Specific comments:

- 1) Distinct transcriptomic clusters of BCR-ABL1 lymphoblastic leukemia
 - I wanted to clarify whether C1 cluster was mainly composed of samples that had Type 1 and Type 2 blasts- if this data is available, and is the case, please mention it in text.
 - Thank you for providing the correlation between % of cells cyMPO+ by flow and bulk RNA-seq MPO gene expression. Does Fig S9e represent normalised counts such as TPM or FPKM?
- 2) Each transcriptomic cluster aligns to a specific stage of normal B-cell development
 - Thank you for mapping the patient samples to relevant ABM sc datasets. The data looks very nice, and I agree it strengthens the manuscript.
 - Thank you for clarifying and amending the B progenitor nomenclature and providing additional gene expression data.
- 3) Distinct transformation events define each molecular subtype
 - Thank you for clarifying that in 3 of the ALLs with Trisomy 21, it is part of the hyperdiploidy. As the authors say, this is expected in hyperdiploid ALL. I asked the question, because these cases should not be represented twice in the figure (3a). It is best to just show the 3 patients with +21 (without hyperdiploidy) in the Trisomy 21 row
- 4) Cellular stage of leukemic transformation impacts treatment outcome in patients
 - If the authors would like to keep Fig 5d, the reason for and limitation of small patient numbers should be clearly discussed in the text

Author Rebuttal, first revision:

REVIEWER #1

We have revised the text of the abstract to address the comment from the Reviewer.

REVIEWER #2

1) In the first section of the Supplementary Results, we show a detailed analysis of blast immunophenotypes, where we show that C1 is enriched for blast types 2 and 3. This is now referenced in the main text in the first Results section.

In figure S9e, MPO gene expression were shown as log2 of raw counts.

2) No response needed.

3) We have added the following sentence to the legend for Fig. 3a. "In hyperdiploid cases, trisomy 21 is a consequence of the hyperdiploid state."

4) We have added the following sentence to the main text. "These survival analyses were limited by small sample sizes due to partitioning of the cohort."

Final Decision Letter:

17th May 2023

Dear Dr. Notta,

I am delighted to say that your manuscript "Transcriptomic classes of BCR-ABL1 lymphoblastic leukemia" has been accepted for publication in an upcoming issue of Nature Genetics.

Your paper will be published online after we receive your corrections and will appear in print in the next available issue. You can find out your date of online publication by contacting the Nature Press Office (press@nature.com) after sending your e-proof corrections. Now is the time to inform your Public Relations or Press Office about your paper, as they might be interested in promoting its publication. This will allow them time to prepare an accurate and satisfactory press release. Include your manuscript tracking number (NG-A60593R1) and the name of the journal, which they will need when they contact our Press Office.

Acceptance is conditional on the data in the manuscript not being published elsewhere, or announced in the print or electronic media, until the embargo/publication date. These restrictions are not intended to deter you from presenting your data at academic meetings and conferences, but any

enquiries from the media about papers not yet scheduled for publication should be referred to us.

Please note that *Nature Genetics* is a Transformative Journal (TJ). Authors may publish their research with us through the traditional subscription access route or make their paper immediately open access through payment of an article-processing charge (APC). Authors will not be required to make a final decision about access to their article until it has been accepted. [Find out more about Transformative Journals](https://www.springernature.com/gp/open-research/transformative-journals)

Authors may need to take specific actions to achieve [compliance](https://www.springernature.com/gp/open-research/funding/policy-compliance-faqs) with funder and institutional open access mandates. If your research is supported by a funder that requires immediate open access (e.g. according to [Plan S principles](https://www.springernature.com/gp/open-research/plan-s-compliance)) then you should select the gold OA route, and we will direct you to the compliant route where possible. For authors selecting the subscription publication route, the journal's standard licensing terms will need to be accepted, including [self-archiving-and-license-to-publish](https://www.nature.com/nature-portfolio/editorial-policies/self-archiving-and-license-to-publish). Those licensing terms will supersede any other terms that the author or any third party may assert apply to any version of the manuscript.

Please note that Nature Portfolio offers an immediate open access option only for papers that were first submitted after 1 January, 2021.

If you have not already done so, we invite you to upload the step-by-step protocols used in this

manuscript to the Protocols Exchange, part of our on-line web resource, natureprotocols.com. If you complete the upload by the time you receive your manuscript proofs, we can insert links in your article that lead directly to the protocol details. Your protocol will be made freely available upon publication of your paper. By participating in natureprotocols.com, you are enabling researchers to more readily reproduce or adapt the methodology you use. Natureprotocols.com is fully searchable, providing your protocols and paper with increased utility and visibility. Please submit your protocol to <https://protocolexchange.researchsquare.com/>. After entering your nature.com username and password you will need to enter your manuscript number (NG-A60593R1). Further information can be found at <https://www.nature.com/nature-portfolio/editorial-policies/reporting-standards#protocols>

Sincerely,

Safia Danovi
Editor
Nature Genetics